# Spike-timing-dependent ensemble encoding by non-classically responsive cortical neurons

Michele N Insanally[1,2,3,4,5], Ioana Carcea[1,2,3,4,5], Rachel E Field[1,2,3,4,5], Chris C Rodgers[6,7], Brian DePasquale[8], Kanaka Rajan[9,10], Michael R DeWeese[11,12], Badr F Albanna[13], Robert C Froemke[1,2,4,5,14]*

[1]Skirball Institute for Biomolecular Medicine, New York University School of Medicine, New York, United States; [2]Neuroscience Institute, New York University School of Medicine, New York, United States; [3]Department of Otolaryngology, New York University School of Medicine, New York, United States; [4]Department of Neuroscience and Physiology, New York University School of Medicine, New York, United States; [5]Center for Neural Science, New York University, New York, United States; [6]Department of Neuroscience, Columbia University, New York, United States; [7]Kavli Institute of Brain Science, Columbia University, New York, United States; [8]Princeton Neuroscience Institute, Princeton University, Princeton, United States; [9]Department of Neuroscience, Icahn School of Medicine at Mount Sinai, New York, United States; [10]Friedman Brain Institute, Icahn School of Medicine at Mount Sinai, New York, United States; [11]Helen Wills Neuroscience Institute, University of California, Berkeley, Berkeley, United States; [12]Department of Physics, University of California, Berkeley, Berkeley, United States; [13]Department of Natural Sciences, Fordham University, New York, United States; [14]Howard Hughes Medical Institute, New York University School of Medicine, New York, United States

*For correspondence:
robert.froemke@med.nyu.edu

**Abstract** Neurons recorded in behaving animals often do not discernibly respond to sensory input and are not overtly task-modulated. These non-classically responsive neurons are difficult to interpret and are typically neglected from analysis, confounding attempts to connect neural activity to perception and behavior. Here, we describe a trial-by-trial, spike-timing-based algorithm to reveal the coding capacities of these neurons in auditory and frontal cortex of behaving rats. Classically responsive and non-classically responsive cells contained significant information about sensory stimuli and behavioral decisions. Stimulus category was more accurately represented in frontal cortex than auditory cortex, via ensembles of non-classically responsive cells coordinating the behavioral meaning of spike timings on correct but not error trials. This unbiased approach allows the contribution of all recorded neurons – particularly those without obvious task-related, trial-averaged firing rate modulation – to be assessed for behavioral relevance on single trials.
DOI: https://doi.org/10.7554/eLife.42409.001

## Introduction

Spike trains recorded from the cerebral cortex of behaving animals can be complex, highly variable from trial-to-trial, and therefore challenging to interpret. A fraction of recorded cells typically exhibit trial-averaged firing rates with obvious task-related features and can be considered 'classically responsive', such as neurons with tonal frequency tuning in the auditory cortex or orientation tuning in the visual cortex. Another population of responsive cells are modulated by multiple task

**eLife digest** Neurons encode information in the form of electrical signals called spikes. Certain neurons increase the rate at which they produce spikes under specific circumstances, e.g., whenever an animal hears a particular sound. These neurons are said to be 'classically responsive'. But not all neurons behave in this way. Others produce spikes at a variable rate that does not obviously relate to the animal's behavior. These neurons are said to be 'non-classically responsive'. They are often omitted from analyses, despite typically outnumbering their classically responsive counterparts. So, what are these neurons doing?

To find out, Insanally et al. trained rats to respond to sounds. The animals learned to poke their nose into a window whenever they heard a specific tone, and to avoid responding whenever they heard any other tone. As the rats performed the task, Insanally et al. recorded from neurons in two areas of the brain, the frontal cortex and the auditory cortex. A computer then analyzed the activity of individual neurons during each trial.

As expected, the firing rate of non-classically responsive cells did not relate to the animals' behavior. But the timing of this firing did. The interval between spikes contained information about which tone the animals had heard and/or how they had responded. The cells worked together in groups to encode this information. Over the course of each trial, every neuron in the group varied the interval between its spikes. Eventually, the group reached a consensus, with all neurons using the same interval to represent information relevant to the task. Groups of neurons in the frontal cortex encoded more information about the category of the tone than those in the auditory cortex.

By including all neurons – both classically and non-classically responsive – this model offers a more comprehensive view of how neural activity relates to behavior. This may in turn help us understand the variable and complex neural activity seen in people with sensory and cognitive disorders.

DOI: https://doi.org/10.7554/eLife.42409.002

parameters ('mixed selectivity cells') and have recently been shown to have computational advantages necessary for flexible behavior (*Rigotti et al., 2013*). However, a substantial number of cells have variable responses that fail to demonstrate firing rates with any obvious trial-averaged relationship to task parameters (*Jaramillo and Zador, 2011*; *Olshausen and Field, 2006*; *Raposo et al., 2014*; *Rodgers and DeWeese, 2014*). These 'non-classically responsive' neurons are especially prevalent in frontal cortical regions but can also be found throughout the brain, including primary sensory cortex (*Hromádka et al., 2008*; *Jaramillo and Zador, 2011*; *Rodgers and DeWeese, 2014*). These response categories are not fixed but can be dynamic, with some cells apparently becoming non-classically responsive during task engagement without impairing behavioral performance (*Carcea et al., 2017*; *Kuchibhotla et al., 2017*; *Otazu et al., 2009*). The potential contribution of these cells to behavior remains to a large extent unknown and represents a major conceptual challenge to the field (*Olshausen and Field, 2006*).

How do these non-classically responsive cells relate to behavioral task variables on single trials? While there are sophisticated approaches for dissecting the precise correlations between classically responsive cells and task structure (*Erlich et al., 2011*; *Jaramillo and Zador, 2011*; *Kiani and Shadlen, 2009*; *Murakami et al., 2014*; *Raposo et al., 2014*), there is still a need for complementary and straightforward analytical tools for understanding any and all activity patterns encountered (*Jaramillo and Zador, 2011*; *Raposo et al., 2014*; *Rigotti et al., 2013*). Moreover, most behavioral tasks produce dynamic activity patterns throughout multiple neural circuits, but we lack unified methods to compare activity across different regions, and to determine to what extent these neurons might individually or collectively perform task-relevant computations. To address these limitations, we devised a novel trial-to-trial analysis using Bayesian inference that evaluates the extent to which relative spike timing in single-unit and ensemble responses encode behavioral task variables.

## Results

### Non-classically responsive cells prevalent in auditory and frontal cortex during behavior

We trained 15 rats on an audiomotor frequency recognition go/no-go task (*Carcea et al., 2017*; *Froemke et al., 2013*; *King et al., 2016*; *Martins and Froemke, 2015*) that required them to nose poke to a single target tone for food reward and withhold from responding to other non-target tones (*Figure 1A*). Tones were 100 ms in duration presented sequentially once every 5–8 s at 70 dB sound pressure level (SPL); the target tone was 4 kHz and non-target tones ranged from 0.5 to 32 kHz separated by one octave intervals. After a few weeks of training, rats had high hit rates to target tones and low false alarm rates to non-targets, leading to high d' values (mean performance shown in *Figure 1B*; each individual rat included in this study shown in *Figure 1—figure supplement 1*).

To correctly perform this task, animals must first recognize the stimulus and then execute an appropriate motor response. We hypothesized that two brain regions important for this behavior are the auditory cortex (AC) and frontal cortical area 2 (FR2). Many but not all auditory cortical neurons respond to pure tones with reliable, short-latency phasic responses (*Hromádka et al., 2008*; *Hubel et al., 1959*; *Kadia and Wang, 2003*; *Merzenich et al., 1975*; *Polley et al., 2007*; *Wehr and Zador, 2003*; *Yaron et al., 2012*). These neurons can process sound in a dynamic and context-sensitive manner, and AC cells are also modulated by expectation, attention, and reward structure, strongly suggesting that AC responses are important for auditory perception and cognition (*David et al., 2012*; *Fritz et al., 2003*; *Hubel et al., 1959*; *Jaramillo and Zador, 2011*; *Weinberger, 2007*). Previously, we found that the go/no-go tone recognition task used here is sensitive to AC neuromodulation and plasticity (*Froemke et al., 2013*). In contrast, FR2 is not thought to be part of the canonical central auditory pathway but is connected to many other cortical regions including AC (*Nelson and Mooney, 2016*; *Schneider et al., 2018*; *Schneider et al., 2014*). This region has recently been shown to be involved in orienting responses, categorization of perceptual stimuli, and in suppressing AC responses during movement (*Erlich et al., 2011*; *Hanks et al., 2015*; *Schneider et al., 2014*). These characteristics suggest that FR2 may be important for goal-oriented behavior.

We first asked if activity in AC or FR2 is required for animals to successfully perform this audiomotor task. We implanted cannulas into AC or FR2 (*Figure 1—figure supplement 2*), and infused the GABA agonist muscimol bilaterally into AC or FR2, to inactivate either region prior to testing behavioral performance. We found that task performance was impaired if either of these regions was inactivated, although general motor functions, including motivation or ability to feed were not impaired (*Figure 1—figure supplement 3*; for AC p=0.03; for FR2 p=0.009 Student's paired two-tailed t-test). Thus activity in both AC and FR2 may be important, perhaps in different ways, for successful performance on this task. We note that a previously published study (*Gimenez et al., 2015*) observed a more modest effect of muscimol-based inactivation of auditory cortex (although we used a separate task and higher dose of muscimol than that study which might contribute to this difference).

Once animals reached behavioral criteria (hit rates $\geq$ 70% and d' values $\geq$ 1.5), they were implanted with tetrode arrays in either AC or FR2 (*Figure 1—figure supplement 4*). After recovery, we made single-unit recordings from individual neurons or small ensembles of two to eight cells during task performance. The trial-averaged responses of some cells exhibited obvious task-related features: neuronal activity was tone-modulated compared to inter-trial baseline activity (*Figure 1C*) or gradually changed over the course of the trial as measured by a ramping index (*Figure 1D*; hereafter referred to as 'ramping activity'). However, 60% of recorded cells were non-classically responsive in that they were neither tone modulated nor ramping according to statistical criteria (*Figure 1E and F*; *Figure 1—figure supplement 5*; 64/103 AC cells and 43/74 FR2 cells from 15 animals had neither significant tone-modulated activity or ramping activity; pre and post-stimulus mean activity compared via subsampled bootstrapping and considered significant when p<0.05; ramping activity measured with linear regression and considered significant via subsampled bootstrapping when p<0.05 and r > 0.5; for overall population statistics see *Figure 1—figure supplement 6*). While the fraction of non-classically responsive AC neurons observed is consistent with previous studies that use different auditory stimuli or behavioral paradigms (*Jaramillo and Zador, 2011*; *Rodgers and DeWeese,*

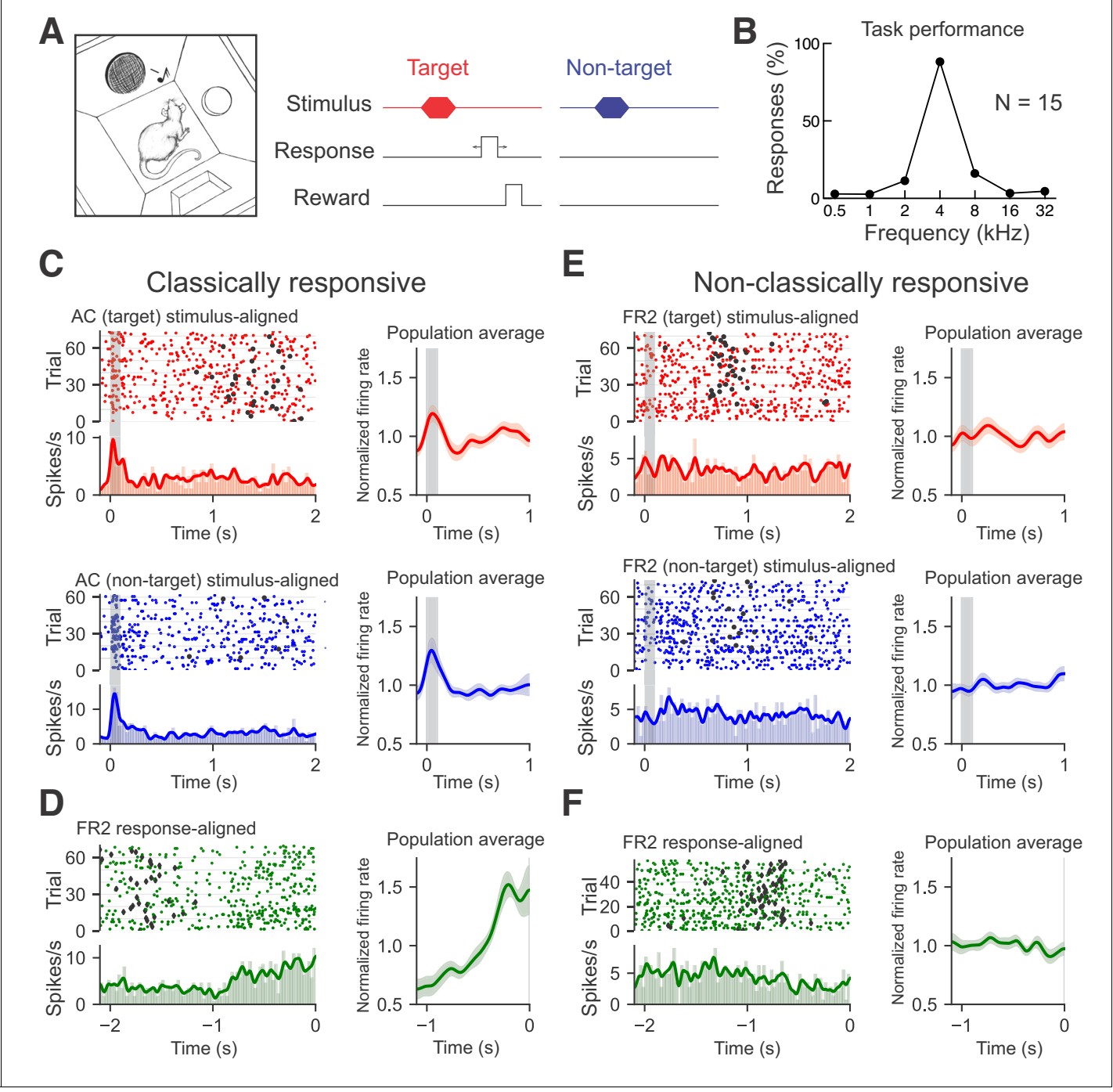

**Figure 1.** Recording from AC or FR2 during go/no-go audiomotor task. (**A**) Behavioral schematic for the go/no-go frequency recognition task. Animals were rewarded with food for entering the nose port within 2.5 s after presentation of a target tone (4 kHz) or given a 7 s time-out if they incorrectly responded to non-target tones (0.5, 1, 2, 8,16, or 32 kHz). (**B**) Behavioral responses (nose pokes) to target and non-target tones (hit rates: 88 ± 7%, false alarms: 7 ± 5%, N = 15 rats). (**C**) Left, AC unit with significant tone modulated responses during target trials (red; top panel, average evoked spikes = 0.55) and non-target trials (blue; bottom panel, average evoked spikes = 0.92). Rasters of individual trials as well as the firing rate histogram and moving average are shown. Histograms of average firing rate during a trial were constructed using 25 ms time bins. A moving average of the firing rate was constructed using a Gaussian kernel with a 20 ms standard deviation. Black circles represent behavioral responses. Right, population averages for all target (n = 23) or nontarget (n = 34) classically responsive singe-units from AC. (**D**) Left, FR2 unit with ramping activity (green; ramp index = 2.82). Trials here are aligned to response time. Diamonds indicate stimulus onset. Right, population average for all ramping single-units from FR2 (n = 21). (**E**) Left, FR2 unit that was not significantly modulated during target trials (red; average evoked spikes = 0.041, p<0.001, 2000 bootstraps). Black circles here represent behavioral responses. Right, population averages for all target (n = 44) or non-target (n = 44) non-classically responsive single-units from FR2

*Figure 1 continued on next page*

*Figure 1 continued*

(F) Left, FR2 unit lacking ramping activity (green, ramp index = −1.0, p<0.001, 2000 bootstraps). Right, population average for all non-ramping single-units from FR2 (n = 44).

DOI: https://doi.org/10.7554/eLife.42409.003

The following figure supplements are available for figure 1:

**Figure supplement 1.** Individual response curves from 15 animals included in this study.

DOI: https://doi.org/10.7554/eLife.42409.004

**Figure supplement 2.** Histological placement of cannulas in AC and FR2.

DOI: https://doi.org/10.7554/eLife.42409.005

**Figure supplement 3.** Bilateral infusion of muscimol into either AC or FR2 significantly impairs task performance.

DOI: https://doi.org/10.7554/eLife.42409.006

**Figure supplement 4.** Histological placement of electrodes in AC and FR2.

DOI: https://doi.org/10.7554/eLife.42409.007

**Figure supplement 5.** Examples of tone-evoked, ramping, and non-classically responsive cells from AC and FR2.

DOI: https://doi.org/10.7554/eLife.42409.008

**Figure supplement 6.** Summary statistics.

DOI: https://doi.org/10.7554/eLife.42409.009

*2014*), this definition does not preclude the possibility that non-classically responsive cells can be driven by other acoustic stimuli or behavioral paradigms.

## Novel single-trial, ISI-based algorithm for decoding non-classically responsive activity

Given that the majority of our recordings were from non-classically responsive cells, we developed a general method for interpreting neural responses even when trial-averaged responses were not obviously task-modulated which allowed us to compare coding schemes across different brain regions (here, AC and FR2). The algorithm is agnostic to the putative function of neurons as well as the task variable of interest (here, stimulus category or behavioral choice).

Our algorithm empirically estimates the interspike interval (ISI) distribution of individual neurons to decode the stimulus category (target or non-target) or behavioral choice (go or no-go) on each trial via Bayesian inference. The ISI was chosen because its distribution could vary between task conditions even without changes in the firing rate – building on previous work demonstrating that the ISI distribution contains complementary information to the firing rate (*Lundstrom and Fairhall, 2006*; *Reich et al., 2000*; *Zuo et al., 2015*). The distinction between the ISI distribution and trial-averaged firing rate is subtle, yet important. While the ISI is obviously closely related to the instantaneous firing rate, decoding with the ISI distribution is not simply a proxy for using the time-varying, trial-averaged rate. To demonstrate this, we constructed three model cells: a stimulus-evoked cell with distinct target and non-target ISI distributions (*Figure 2A*), a stimulus-evoked cell with identical ISI distributions (*Figure 2B*), and a non-classically responsive cell with distinct target and non-target ISI distributions (*Figure 2C*). These models clearly demonstrate that trial-averaged rate modulation can occur with or without corresponding differences in the ISI distributions and cells without apparent trial-averaged rate-modulation can nevertheless have distinct ISI distributions. Taken together, these examples demonstrate that the ISI distribution and trial-averaged firing rate capture different spike train statistics. This has important implications for decoding non-classically responsive cells that by definition do not exhibit large firing rate modulations but nevertheless may contain information latent in their ISI distributions.

For each recorded neuron, we built a library of ISIs observed during target trials and a library for non-target trials from a set of 'training trials'. Two different cells from AC are shown in *Figure 3A* and *Figure 3—figure supplement 1A–D*, and another cell from FR2 is shown in *Figure 3—figure supplement 1E–H*. These libraries were used to infer the probability of observing an ISI during a particular trial type (*Figure 3B,C*; *Figure 3—figure supplement 1C,G*; left panels show target in red and non-target in blue). These conditional probabilities were inferred using non-parametric statistical methods to minimize assumptions about the underlying process generating the ISI distribution and better capture the heterogeneity of the observed ISI distributions (*Figure 3B*; *Figure 3—figure supplement 1C,G*). We verified that our observed distributions were better modeled by non-parametric

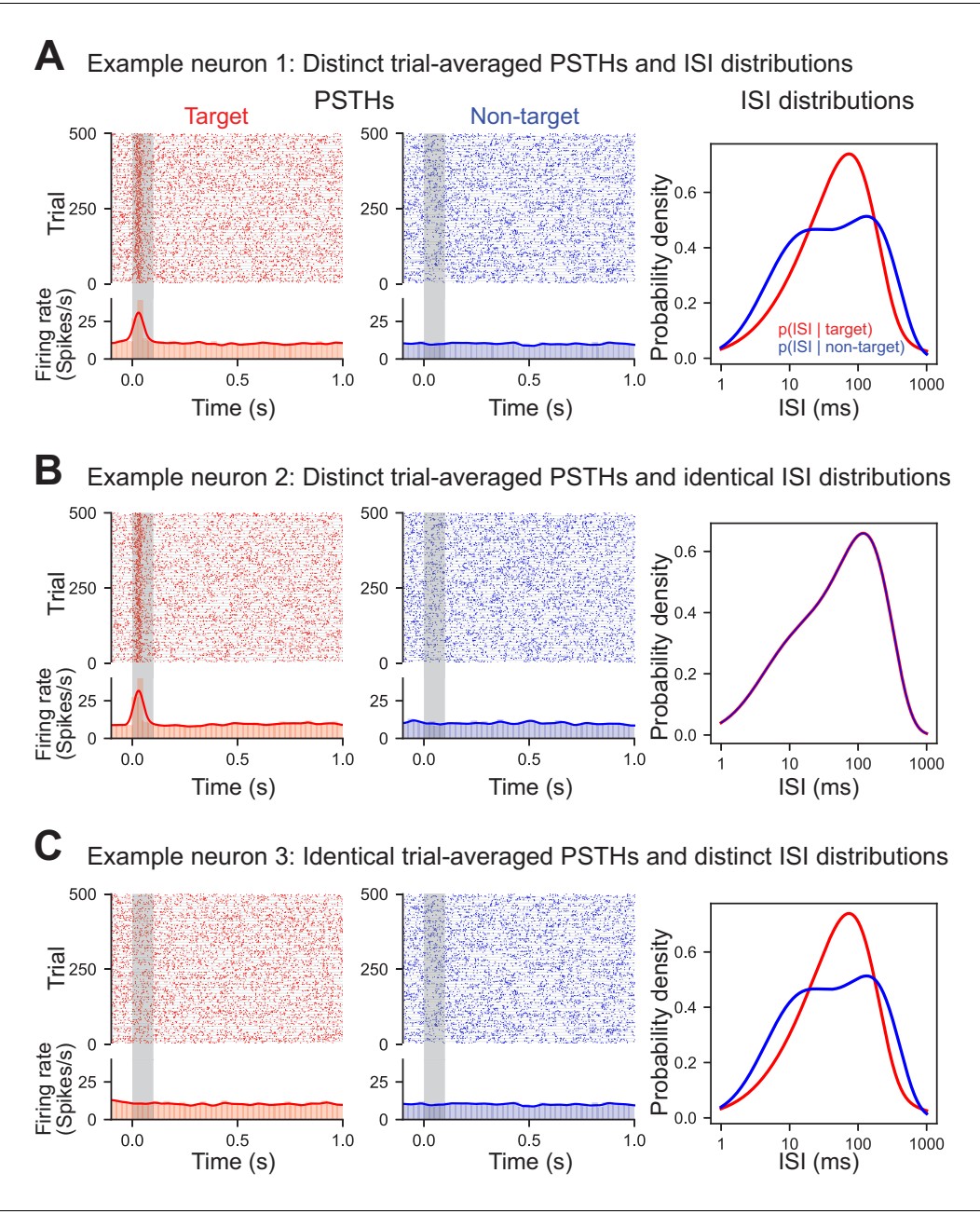

**Figure 2.** ISIs capture information distinct from trial-averaged rate. Three simulated example neurons demonstrating that differences in the ISI are not necessary for differences in the trial-averaged firing rate to occur (and vice versa). Each trial was generated by randomly sampling from the appropriate conditional ISI distribution. Evoked responses were generated by shifting trials without altering the ISI distributions such that one spike during stimulus presentation is found at approximately 30 ms (with a variance of 10 ms). (**A**) Example neuron with both an evoked target response and a difference in the conditional ISI distributions on target and non-target trials. (**B**) Example neuron with an evoked target response but identical conditional ISI distributions. (**C**) Example non-classically responsive neuron with no distinct trial-averaged activity relative to the pre-stimulus period that nevertheless is generated by distinct ISI distributions.

DOI: https://doi.org/10.7554/eLife.42409.010

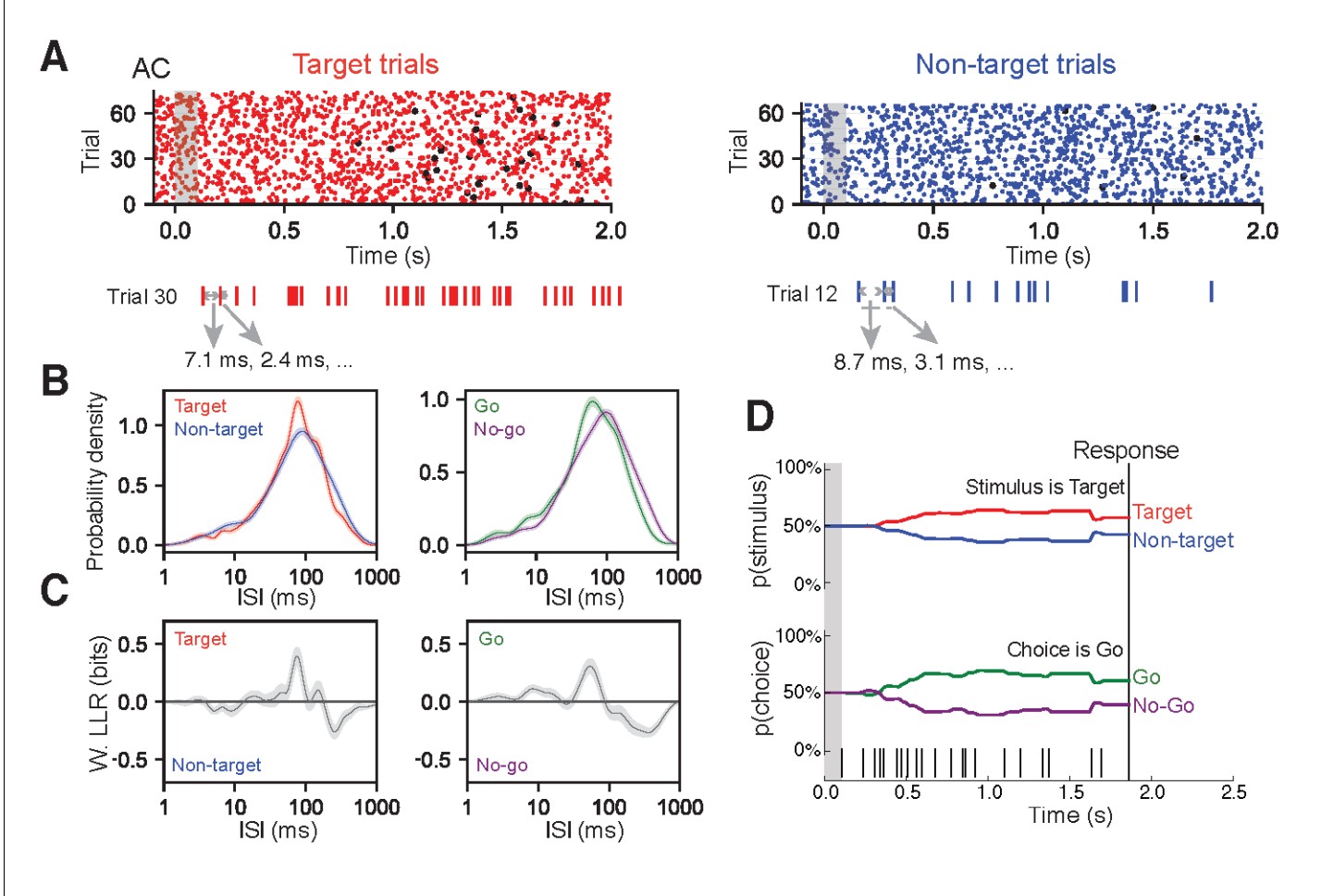

**Figure 3.** ISI-based algorithm for decoding behavioral variables from AC and FR2 single-units. (**A**) Single-unit activity was first sorted by task condition, here for target trials (red) and non-target trials (blue). All ISIs following stimulus onset and before behavioral choice were aggregated into libraries for each condition (average response time is used on no-go trials) as shown for a sample trial. (**B**) Probability density for observing a given ISI on each condition was generated via Kernel Density Estimation on libraries from (**A**) Left, target (red) and non-target (blue) probabilities. Right, go (green) and no-go (purple). (**C**) Relative differences between the two stimulus conditions (or choice conditions) was used to infer the actual stimulus category (or choice) from an observed spike train, in terms of weighted log likelihood ratio (W. LLR) for stimulus category ($p(ISI)*(log_2 p(ISI|target) - log_2(ISI|non-target))$; on left) and behavioral choice ($p(ISI)*(log\ p_2(ISI|go) - log_2(ISI|no-go))$; on right). When curve is above zero the ISI suggests target (go) and when below zero the ISI suggests non-target (no-go). (**D**) Probability functions from B were used as the likelihood function to estimate the prediction of a spike train on an individual trial (bottom). Bayes' rule was used to update the probability of a stimulus (top) or choice (bottom) as the trial progressed and more ISIs were observed. The prediction for the trial was assessed at the end of the trial as the probability of stimulus category (or choice). In this example trial, p(target|ISI)=61%.

DOI: https://doi.org/10.7554/eLife.42409.011

The following figure supplements are available for figure 3:

**Figure supplement 1.** Decoding algorithm to determine stimulus category and choice in single-unit ISIs from AC and FR2 for two additional neurons.
DOI: https://doi.org/10.7554/eLife.42409.012
**Figure supplement 2.** Empirical ISI distributions are better modeled using non-parametric methods.
DOI: https://doi.org/10.7554/eLife.42409.013

methods rather than standard parametric methods (e.g. rate-modulated Poisson process; *Figure 3—figure supplement 2*). Specifically, we found the distributions using Kernel Density Estimation where the kernel bandwidth for each distribution was set using 10-fold cross-validation. To accommodate any non-stationarity, these ISI distributions were calculated in 1 s long sliding windows recalculated every 100 ms over the course of the trial. We then used these training set probability functions to decode a spike train from a previously unexamined individual trial from the set of remaining 'test

trials'. This process was repeated 124 times using 10-fold cross-validation with randomly generated folds.

Importantly, while the probabilities of observing particular ISIs on target and non-target trials were similar (*Figure 3B*; *Figure 3—figure supplement 1C,G*), small differences between the curves carried sufficient information to allow for decoding. To characterize these differences, we used the weighted log likelihood ratio (W. LLR; *Figure 3C*; *Figure 3—figure supplement 1C,G*) to clearly represent which ISIs suggested target (W. LLR > 0) or non-target (W. LLR < 0) stimulus categories. Our algorithm relies only on statistical differences between task conditions; therefore, the W. LLR summarizes all spike timing information necessary for decoding. Similar ISI libraries were also computed for behavioral choice categories (*Figure 3B,C*; *Figure 3—figure supplement 1C*; right panels show go decision in green and no-go in purple). These examples clearly illustrate that the relationship between the ISIs and task variables cannot simply be approximated by an ISI or firing rate threshold where short ISIs imply one task variable and longer ISIs imply another: in the cell shown in *Figure 3*, short ISIs (ISI < 50 ms) indicated non-target, medium ISIs (50 ms <ISI < 100 ms) indicated target, and longer ISIs indicated non-target (100 ms <ISI).

The algorithm uses the statistical prevalence of certain ISI values under particular task conditions (in this case the ISIs accompanying stimulus category or behavioral choice), to infer the task condition for each trial. Each trial begins with equally uncertain probabilities about the stimulus categories (i.e. p(target)=p(non-target)=50%). As each ISI is observed sequentially within the trial, the algorithm applies Bayes' rule to update p(target|ISI) and p(non-target|ISI) using the likelihood of the ISI under each stimulus category (p(ISI|target) and p(ISI|non-target) (*Figure 3B–D*). As these functions were estimated in 1 s long sliding windows, each ISI was assessed using the distribution that placed the final spike closest to the center of the sliding window. As shown for one trial of the example cell in *Figure 3D*, ISIs observed between 0 and 1.0 s consistently suggested the presence of the target tone, whereas ISIs observed between 1.0 and 1.4 s suggested the non-target category thereby also necessarily reducing the belief that a target tone was played (*Figure 3D*, top trace). These ISI likelihood functions consider each ISI to be independent of previous ISIs and therefore ignore correlations between ISIs. After this process was completed for all ISIs in the particular trial, we obtained the probability of a non-target tone and a target tone as a function of time during the trial (*Figure 3D*). Because it is particularly challenging to dissociate choice from motor execution or preparatory motor activity in this task paradigm, the prediction for the entire trial p(target|ISI) is evaluated at the end of the trial (in the example trial, p(target|ISI)=61%; *Figure 3D*). This process is repeated for the behavioral choice (*Figure 3B–D*; right panels; trials separated according to go, no-go; probabilities of ISIs in each condition generated; conditional probabilities used as likelihood function to predict behavioral choice on a given trial). The single-trial decoding performance of each neuron is then averaged over all trials as a measure of the overall ability of each neuron to distinguish behavioral conditions (*Figure 4A*). Note that this measure not only takes into account whether the algorithm was correct on individual trials (i.e. target vs. non-target), but also its prediction certainty.

## Non-classically responsive cells contain spike-timing-based task information

Can we uncover task information from non-classically responsive cells? We found that non-classically responsive cells in both AC and FR2 provided significant spike-timing-based information about each task variable (*Figure 4A,B*, red; *Figure 4—figure supplement 1*). The ability to decode was poorly explained by the average firing rate (*Figure 4—figure supplements 2A–F* and 0.30 < r < 0.46), z-score (*Figure 4—figure supplements 2G–I*, –0.05 < r < 0.05), and ramping activity (*Figure 4—figure supplements 2J*, –0.02 < r < 0.28). Stimulus decoding performance was also independent of receptive field properties including best frequency and tuning curve bandwidth for AC neurons (*Figure 4—figure supplement 3*).

We also observed that task information was distributed across both AC and FR2, and neural spike trains from individual units were multiplexed in that they often encoded information about both stimulus category and choice simultaneously (*Figure 4B*, *Table 1*). Given the strong correlation between stimulus and choice variables in the task design, it is difficult to fully separate information about one variable from information about the other. To establish that multiplexing was not simply a byproduct of this correlation, an independent measure of multiplexing relying on multiple regression

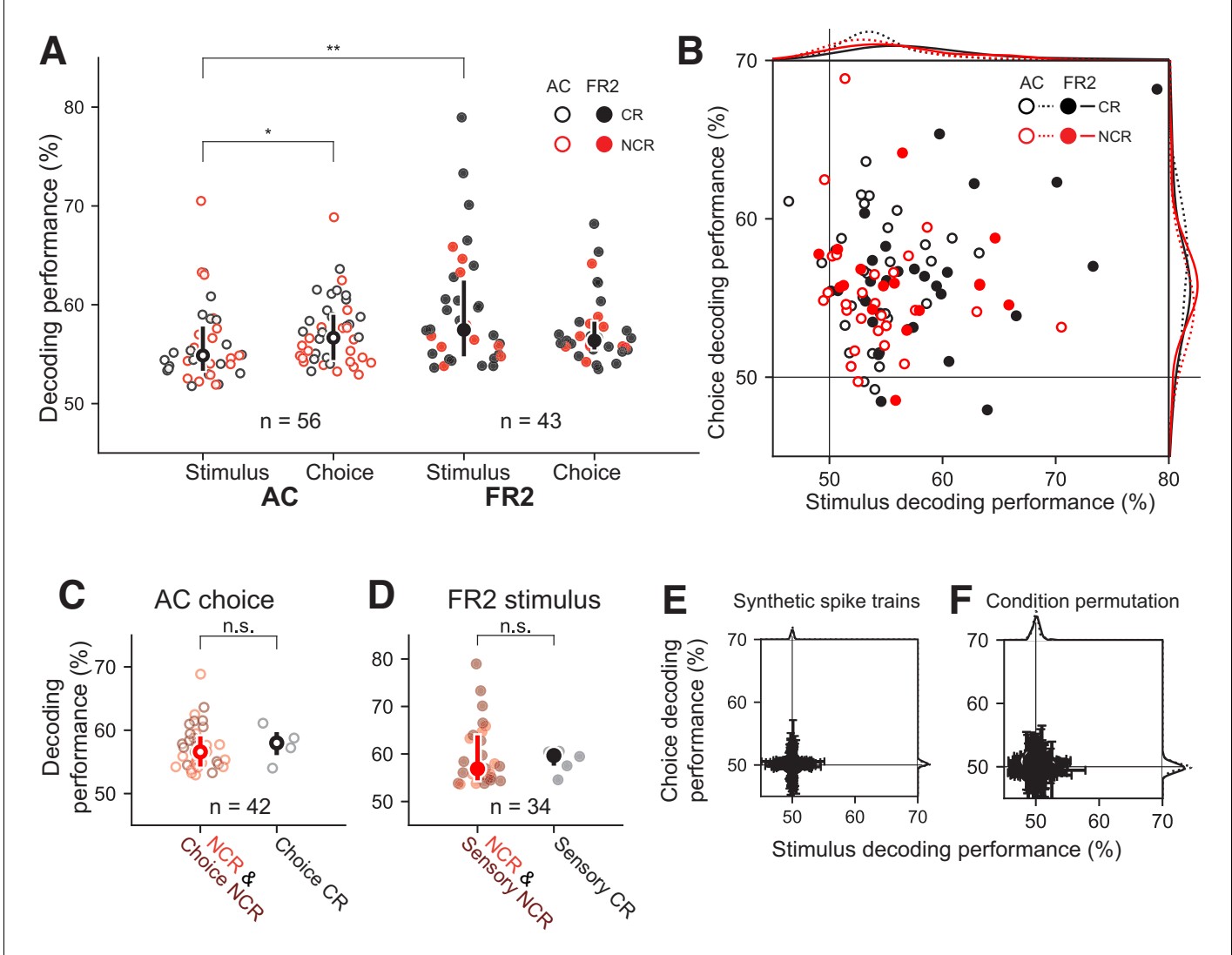

**Figure 4.** Decoding performance of single-units recorded from AC or FR2. (**A**) Decoding performance of single-units for stimulus category and behavioral choice in AC (open circles) and FR2 (filled circles) restricted to those statistically significant relative to synthetically-generated spike trains (p<0.05, permutation test, two-sided). Note that decoding performance values reflect the algorithm's prediction certainty on individual trials. Central symbol with error bars represents group medians and top and bottom quartiles (*p=0.02, **p=0.001, Mann-Whitney U test, two-sided). Black symbols, classically responsive cells; red symbols, non-classically responsive cells. (**B**) Weighted decoding performance for choice versus stimulus, restricted to those statistically significant relative to synthetically generated spike trains for either stimulus, choice, or both (p<0.05, permutation test, two-sided). Black symbols, classically responsive cells; red symbols, non-classically responsive cells. This performance metric is weighted by the prediction certainty on individual trials and can be thought of as a confidence measure. (**C**) Choice decoding performance in AC of non-classically responsive cells (red) and choice non-classically responsive (dark-red) versus choice classically responsive cells (black; i.e. ramping cells). Decoding performance was not statistically different (p=0.32 Mann-Whitney U test, two-sided). Central symbol with error bars represents group medians and top and bottom quartiles. (**D**) Stimulus decoding performance in FR2 for non-classically responsive cells (red) and sensory non-classically responsive (dark-red) versus choice responsive cells (black; i.e. ramping cells). Decoding performance was not statistically different (p=0.29, Mann-Whitney U test, two-sided). Central symbol with error bars represents group medians and top and bottom quartiles. (**E**) Decoding performance for choice versus stimulus, applied to spike trains synthetically generated from sampling (with replacement) over all ISIs observed without regard to stimulus category or behavioral choice. Black, classically responsive cells; red, non-classically responsive cells. Error bars represent standard deviation. (**F**) Decoding performance for choice versus stimulus, applied to spike trains left intact but trial conditions (stimulus category and behavioral choice) were randomly permuted (1000 permutations per unit). Error bars represent standard deviation.

DOI: https://doi.org/10.7554/eLife.42409.014

The following figure supplements are available for figure 4:

**Figure supplement 1.** Decoding performance of single cells on individual trials.

DOI: https://doi.org/10.7554/eLife.42409.015

*Figure 4 continued on next page*

*Figure 4 continued*

**Figure supplement 2.** Lack of correlations between classical firing rate metrics and stimulus or choice decoding performance.
DOI: https://doi.org/10.7554/eLife.42409.016
**Figure supplement 3.** Stimulus decoding in AC independent of receptive field properties.
DOI: https://doi.org/10.7554/eLife.42409.017
**Figure supplement 4.** Decoding performance is a sufficient measure of uni/multiplexing.
DOI: https://doi.org/10.7554/eLife.42409.018

was applied (*Figure 4—figure supplement 4*). This analysis confirmed that the information revealed by our algorithm about a behavioral variable was primarily a reflection of that variable and not simply an indirect measure of the other, correlated variable. This analysis establishes that a certain degree of separability is possible and demonstrates that the multiplexing observed in our decoding results is unlikely to be a trivial byproduct of correlations in the task variables.

Despite the broad sharing of information about behavioral conditions, there were notable systematic differences between AC and FR2. Surprisingly, neurons in FR2 were more informative about stimulus category than AC, and AC neurons were more informative about choice than stimulus category (*Figure 4A*, $p_{AC}$ = 0.016, $p_{stim}$ = 0.0013, Mann-Whitney U test, two-sided). Both of these observations would not have been detected at the level of the PSTH, as most cells in AC were non-classically responsive for behavioral choice (no ramping activity, 91/103), yet our decoder revealed that these same cells were as informative as choice classically responsive cells (*Figure 4C*, p=0.32 Mann-Whitney U test, two-sided; red circles indicate cells non-classically responsive for both variables, dark-red cells are choice non-classically responsive, and black cells are classically responsive). Similarly, most cells in FR2 were sensory non-classically responsive (not tone modulated, 60/74), yet contained comparable stimulus information to sensory classically responsive cells (*Figure 4D*, p=0.29 Mann-Whitney U test, two-sided; red cells are non-classically responsive for both variables, dark-red cells are sensory non-classically responsive, black cells are classically responsive).

To assess the statistical significance of these results, we tested our algorithm on two shuffled data sets. First, we ran our analysis using synthetically-generated trials that preserved trial length but randomly sampled ISIs with replacement from those observed during a session without regard to condition (*Figure 4E*). Second, we left trial activity intact but permuted the stimulus category and choice for each trial (*Figure 4F*). We restricted analysis to cells with decoding performance significantly different from synthetic spike trains (all cells in *Figure 4A–D* significantly different from synthetic condition shown in *Figure 4E*, p<0.05, bootstrapped 1240 times).

To directly assess the extent to which information captured by the ISI distributions in our data set was distinct from the time-varying rate, we compared the performance from our ISI-based decoder to a conventional rate-modulated (inhomogeneous) Poisson decoder (*Rieke et al., 1999*), which assumes that spikes are produced randomly with an instantaneous probability equal to the time-varying firing rate. As our model cells illustrate (*Figure 2*), it is possible to decode using the ISI distributions even when firing rates are uninformative (*Figure 5A*). When applied to our dataset, the ISI-based decoder generally outperformed this conventional rate-based decoder confirming that ISIs capture information distinct from that of the firing rate (*Figure 5B*; Overall stimulus decoding performance: $p_{AC}$ = 0.0001, $p_{FR2}$ = 8 × 10$^{-6}$; Overall choice decoding performance $p_{AC}$ = 0.0057, $p_{FR2}$ = 0.02, Mann-Whitney U test, two-sided). Moreover, comparing single trial decoding outcomes demonstrated weak to no correlations between the ISI-based decoder and the conventional rate

**Table 1.** Number of classically responsive (CR) or non-classically responsive (NCR) neurons in AC and FR2 with significant stimulus or choice information.

|  |  | # stimulus sig. | # choice sig. | Total # |
|---|---|---|---|---|
| AC | CR | 19 | 21 | 39 |
|  | NCR | 18 | 21 | 64 |
| FR2 | CR | 20 | 22 | 31 |
|  | NCR | 10 | 11 | 43 |

DOI: https://doi.org/10.7554/eLife.42409.019

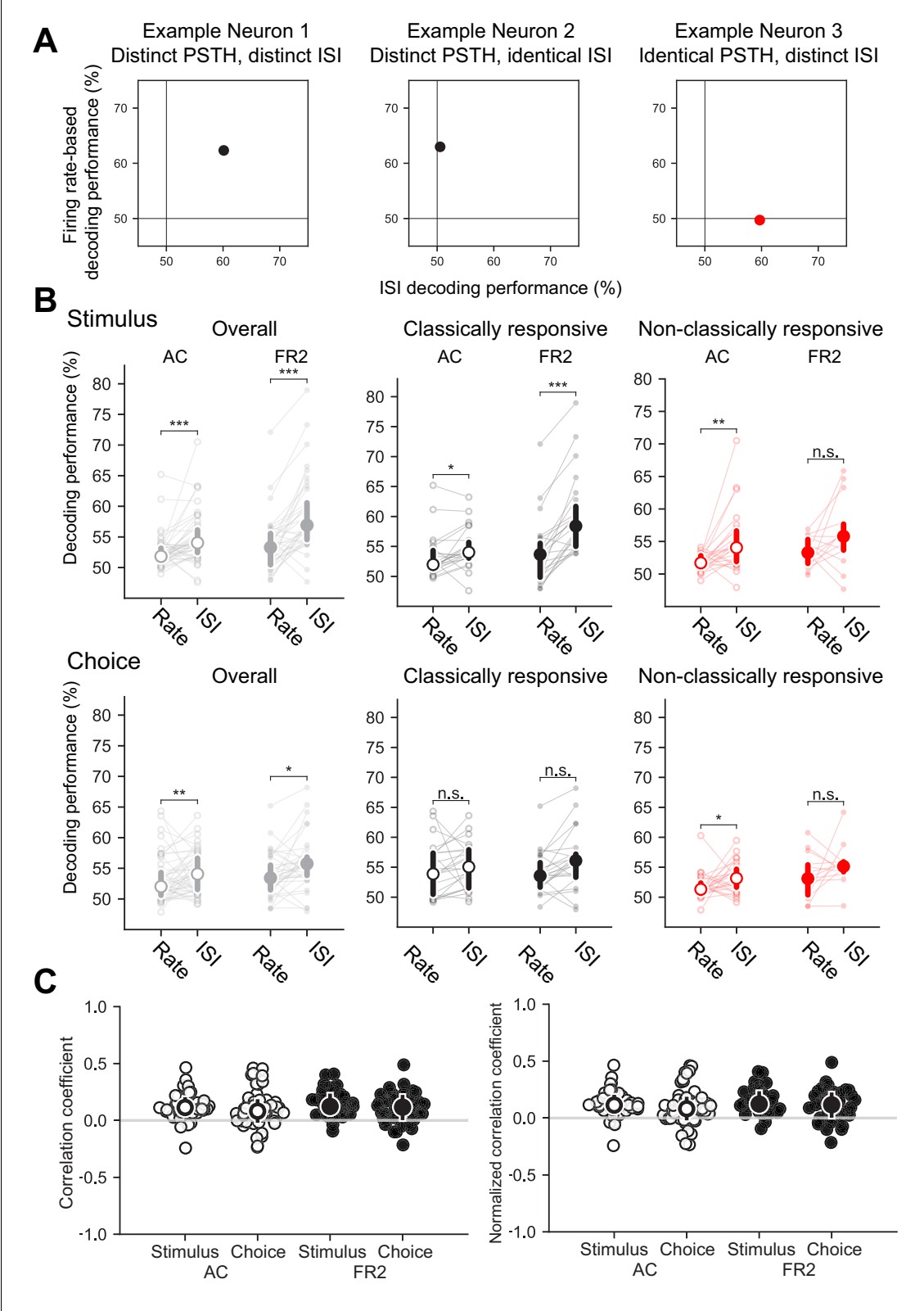

**Figure 5.** Information captured by ISI-based decoder distinct from conventional rate-modulated (inhomogeneous) Poisson decoder. (**A**) Decoding performance comparison for example neurons shown in *Figure 2*. *Left*, Both the trial-averaged firing rate and the ISI distributions can be used to decode stimulus category for this example neuron. *Middle*, Only the firing rate can be used to decode this example. *Right*, In this case, the ISI distributions can be used to decode even when the trial-averaged firing rate cannot. (**B**) Comparison of decoding performance for conventional rate-

*Figure 5 continued on next page*

*Figure 5 continued*

modulated decoder to our ISI-based decoder. *Top row*, stimulus decoding, *bottom row*, choice decoding. *Left*, Overall comparison for all cells. *Right*, Comparison for classically responsive and non-classically responsive cells (Stimulus Overall: ***$p_{AC}$ = 0.0001, ***$p_{FR2}$ = 8 × 10$^{-6}$, Stimulus Repsonsive: *$p_{AC}$ = 0.031, ***$p_{FR2}$ = 4 × 10$^{-5}$, Stimulus non-classically responsive: **$p_{AC}$ = 0.0019, n.s. $p_{FR2}$ = 0.096, Choice Overall: **$p_{AC}$ = 0.0057, *$p_{FR2}$ = 0.02, Choice Repsonsive: n.s. $p_{AC}$ = 0.031, n.s. $p_{FR2}$ = 0.08, Choice non-classically responsive: *$p_{AC}$ = 0.004, n.s. $p_{FR2}$ = 0.19, Wilcoxon signed-rank test). Individual cells shown and median with error bars designating bottom and top quartiles superimposed. (C) *Left*, Matthews correlation coefficient (MCC) between correct predictions of our ISI-based decoder and a conventional rate-modulated firing rate decoder. A MCC value of 1 indicates each decoder correctly decodes exactly the same set of trials, whereas −1 indicates each decoder is correct on complementary trials. Values close to 0 indicate that that the relationship between the decoders is close to chance. Typically, values from −0.5 to 0.5 are considered evidence for weak to no correlation (stimulus median and interquartile range: AC = 0.10, 0.09, FR2 = 0.11, 0.12; choice median and interquartile range: AC = 0.06, 0.15, FR2 = 0.08, 0.17). *Right*, Matthews correlation coefficient (MCC) rescaled by the maximum possible correlation given the decoding performance of each method remains fixed. This control demonstrates that the correlation values are not a result of weak decoding performance for one of the decoding methods (stimulus median and interquartile range: AC = 0.11, 0.11, FR2 = 0.12, 0.15; choice median and interquartile range: AC = 0.08, 0.17, FR2 = 0.11, 0.19). Source data has been provided in the spreadsheet titled 'figure_5.csv'.

DOI: https://doi.org/10.7554/eLife.42409.020

The following figure supplement is available for figure 5:

**Figure supplement 1.** Whole-cell recordings from AC and FR2 neurons showing that different cells can have distinct responses to the same input pattern- necessary for ISI-based decoding by biological networks.

DOI: https://doi.org/10.7554/eLife.42409.021

decoder, further underscoring that these two methods rely on different features of the spike train to decode (*Figure 5C*; stimulus medians: AC = 0.10 FR2 = 0.11; choice medians: AC = 0.07, FR2 = 0.08).

We hypothesize that ISI-based decoding is biologically plausible. Short-term synaptic plasticity and synaptic integration provide powerful mechanisms for differential and specific spike-timing-based coding. We illustrated this capacity by making whole-cell recordings from AC neurons in vivo and in brain slices (*Figure 5—figure supplement 1A,B*), as well as in FR2 brain slices (*Figure 5—figure supplement 1C*). In each case, different cells could have distinct response profiles to the same input pattern, with similar overall rates but different spike timings.

Moreover, we note that this type of coding scheme requires few assumptions about implementation, and does not require additional separate integrative processes to compute rates or form generative models. Thus, ISI-based decoding coding could be generally applicable across brain areas, as demonstrated here for AC and FR2.

## Non-classically responsive cells encode selection rule information in a novel task-switching paradigm

To further demonstrate the generalizability and utility of our approach, we applied our decoding algorithm to neurons that were found to be non-classically responsive in a previously published study (*Rodgers and DeWeese, 2014*). In this study, rats were trained on a novel auditory stimulus selection task where depending on the context animals had to respond to one of two cues while ignoring the other. Rats were presented with two simultaneous sounds (a white noise burst and a warble). In the 'localization' context, the animal was trained to ignore the warble and respond to the location of the white noise burst and in the 'pitch' context it was trained to ignore the location of the white noise burst and respond to the pitch of the warble (*Figure 6A*). Using our algorithm, we found significant stimulus and choice-related information in the activity of non-classically responsive cells that displayed no stimulus modulation nor ramping activity in the firing rate (*Figure 6B–D*). The main finding of the study is that the pre-stimulus activity in both primary auditory cortex and prefrontal cortex encodes the selection rule (i.e. activity reflects whether the animal is in the localization or pitch context). This conclusion was entirely based on a difference in pre-stimulus firing rate between the two contexts. The authors reported, but did not further analyze, cells that did not modulate their pre-stimulus firing rate. In our nomenclature these cells are 'non-classically responsive for the selection rule'. Using our algorithm, we found that the ISI distributions of these cells encoded the selection rule and were significantly more informative than the classically responsive cells (*Figure 6E*, $p_{AC}$ = 5 × 10$^{-6}$, $p_{PFC}$ <0.0002, Mann-Whitney U test, two-sided). This surprising result demonstrates that our algorithm generalizes to novel datasets, and may be used to uncover coding for cognitive

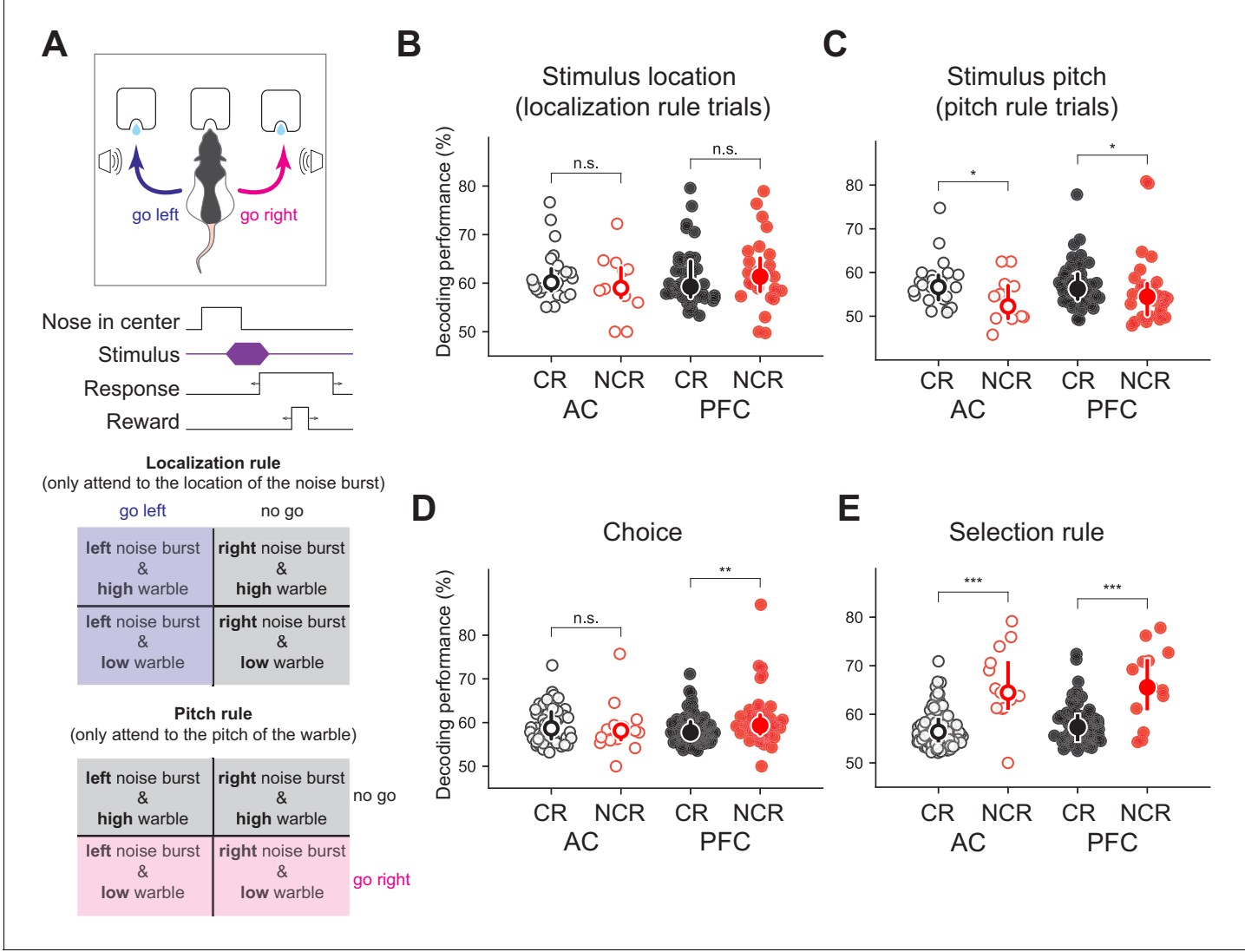

**Figure 6.** Non-classically responsive cells in both auditory cortex and prefrontal cortex (PFC) encode behavioral variables including the selection rule in a task switching paradigm. (A) Schematic of novel auditory stimulus selection task. Animals were presented with two simultaneous tones (a white noise burst and warble) and trained to respond to the location of the sound in the 'localization' context while ignoring pitch and respond to the pitch while ignoring the location in the 'pitch' context (figure adapted from *Rodgers and DeWeese, 2014*), *Neuron*). Decoding performance for (B) stimulus localization on localization trials ($p_{AC} = 0.24$, $p_{PFC} = 0.21$, Mann-Whitney U test, two-sided), (C) stimulus pitch on pitch trials ($p_{AC} = 0.48$, $p_{PFC} = 0.47$, Mann-Whitney U test, two-sided), and (D) choice (**$p_{AC} = 0.0064$, $p_{PFC} = 0.22$, Mann-Whitney U test, two-sided) for classically responsive cells (black) and non-classically responsive cells (red; no stimulus modulation or ramping activity) in auditory (open symbols) and prefrontal cortex (closed symbols) previously reported but not further analyzed in this study. (E) Decoding performance for the selection rule for classically responsive (black) and non-classically responsive cells (red; similar pre-stimulus firing rates for both pitch and localization blocks; ***$p_{AC} = 5 \times 10^{-6}$, ***$p_{PFC} < 0.0002$, Mann-Whitney U test, two-sided).

DOI: https://doi.org/10.7554/eLife.42409.022

variables beyond those apparent from conventional trial-averaged, rate-based analyses. Furthermore, these results indicate that as task complexity increases non-classically responsive cells are differentially recruited for successful task execution.

## Non-classically responsive ensembles are better predictors of behavioral errors

Downstream brain regions must integrate the activity of many neurons and this ISI-based approach naturally extends to simultaneously recorded ensembles. We therefore asked whether using small

ensembles would change or improve decoding. To decode from ensembles, likelihood functions from each cell were calculated independently as before but were used to simultaneously update the task condition probabilities (p(target | ISI) and p(go | ISI)) on each trial (*Figure 7A*). Analyzing ensembles of two to eight neurons in AC and FR2 significantly improved decoding for both variables in FR2 and stimulus decoding in AC (*Figure 7B*, $p_{AC\ stim}$=0.04, $p_{FR2\ stim}$=$1\times10^{-5}$, $p_{AC}$ = 0.29, $p_{FR2\ choice}$=$7\times10^{-5}$, Mann-Whitney U test, two-sided). This was not a trivial consequence of using more neurons, as the information provided by individual ISIs on single trials can be contradictory (e.g. compare LLR functions in *Figure 3C* and *Figure 3—figure supplement 1C* for 50 ms <ISIs < 120 ms). For ensemble decoding to improve upon single neuron decoding, the ISIs of each member of the ensemble must indicate the same task variable.

Can our decoding method predict errors on a trial-by-trial basis? In general, trial-averaged PSTHs did not reveal systematic differences between correct and error trials (*Figure 7—figure supplement 1*). However, when we examined single-trial performance with our algorithm, ensembles of neurons in AC and FR2 predicted behavioral errors (*Figure 7C*). In general, ensembles in AC predicted behavioral errors significantly better than those in FR2 (*Figure 7C*, for three-member ensembles: p=$1.2 \times 10^{-5}$, for four-member ensembles: p=0.03, Mann-Whitney U test, two-sided). Interestingly, decoding with an increasing number of non-classically responsive cells improved error prediction in both AC and FR2 (*Figure 7D*, ($p_{AC}$ = 0.013, $p_{FR2}$ = 0.046, Welch's t-test).

## Timing-dependent ensemble consensus-building dynamics underlie task information

While improvements were seen in decoding performance with increasing ensemble size, the ISI distributions/ISI-based likelihood functions were highly variable across individual ensemble members. Thus, we wondered if there was task-related structure in the timing of population activity that evolved over the course of the trial to instantiate behavior. To answer this question, we examined whether local ensembles share the same representation of task variables over the course of the trial. Do they 'reach consensus' on how to represent task variables using the ISI (*Figure 8A*)? Without consensus, a downstream area would need to interpret ensemble activity using multiple disparate representations rather than one unified code (*Figure 8B*). The firing rates and ISI distributions of simultaneously-recorded units were generally variable across cells requiring an exploratory approach to answer this question (*Figure 8C*, example three-member ensemble with heterogeneous conditional ISI distributions). Therefore, we examined changes in the distributions of ISIs across task conditions, asking how the moment-to-moment changes in the log-likelihood ratio (LLR) of each cell were coordinated to encode task variables (*Figure 8C*). We focused on the LLR because it quantifies how the ISI represents task variables for a given cell and summarizes all spike timing information needed by our algorithm (or a hypothetical downstream cell) to decode.

We examined how ensembles coordinate their activity moment-to-moment over the course of the trial by quantifying the similarity of the LLRs across cells in a sliding window. Similarity was assessed by summing the LLRs of ensemble members, calculating the total area underneath the resulting curve, and normalizing this value by the sum of the areas of each individual LLR. We refer to this quantified similarity as 'consensus'; a high consensus value indicates that ensemble members have similar LLRs and therefore have a similar representation of task variables (*Figure 8D*). We should emphasize that successful ensemble decoding (*Figure 7*) does not require the LLRs of ensemble members to be related in any way; therefore, structured LLR dynamics (*Figure 8*) are not simply a consequence of how our algorithm is constructed.

While the conventional trial-averaged PSTH of non-classically responsive ensembles recorded in AC and FR2 showed no task-related modulation, our analysis revealed structured temporal dynamics of the LLRs (captured by the consensus value). On correct trials, we observe a trajectory of increasing consensus at specific moments during the trial signifying a dynamically created, shared ISI representation of task variables. In FR2, sensory non-classically responsive ensembles (ensembles in which at least two out of three cells were not tone-modulated) encode stimulus information using temporally-precise stimulus-related dynamics on correct trials. The stimulus representation of sensory non-classically responsive ensembles reached consensus rapidly after stimulus onset followed by divergence (*Figure 8E*, stimulus-aligned, solid line, Δconsensus, t = 0 to 0.42 s, $p_{SNR}$ = $3.9 \times 10^{-4}$ Wilcoxon test with Bonferroni correction, two-sided). Sensory classically responsive ensembles in AC increased consensus beyond stimulus presentation, reaching a maximum ~750 ms after tone onset on correct

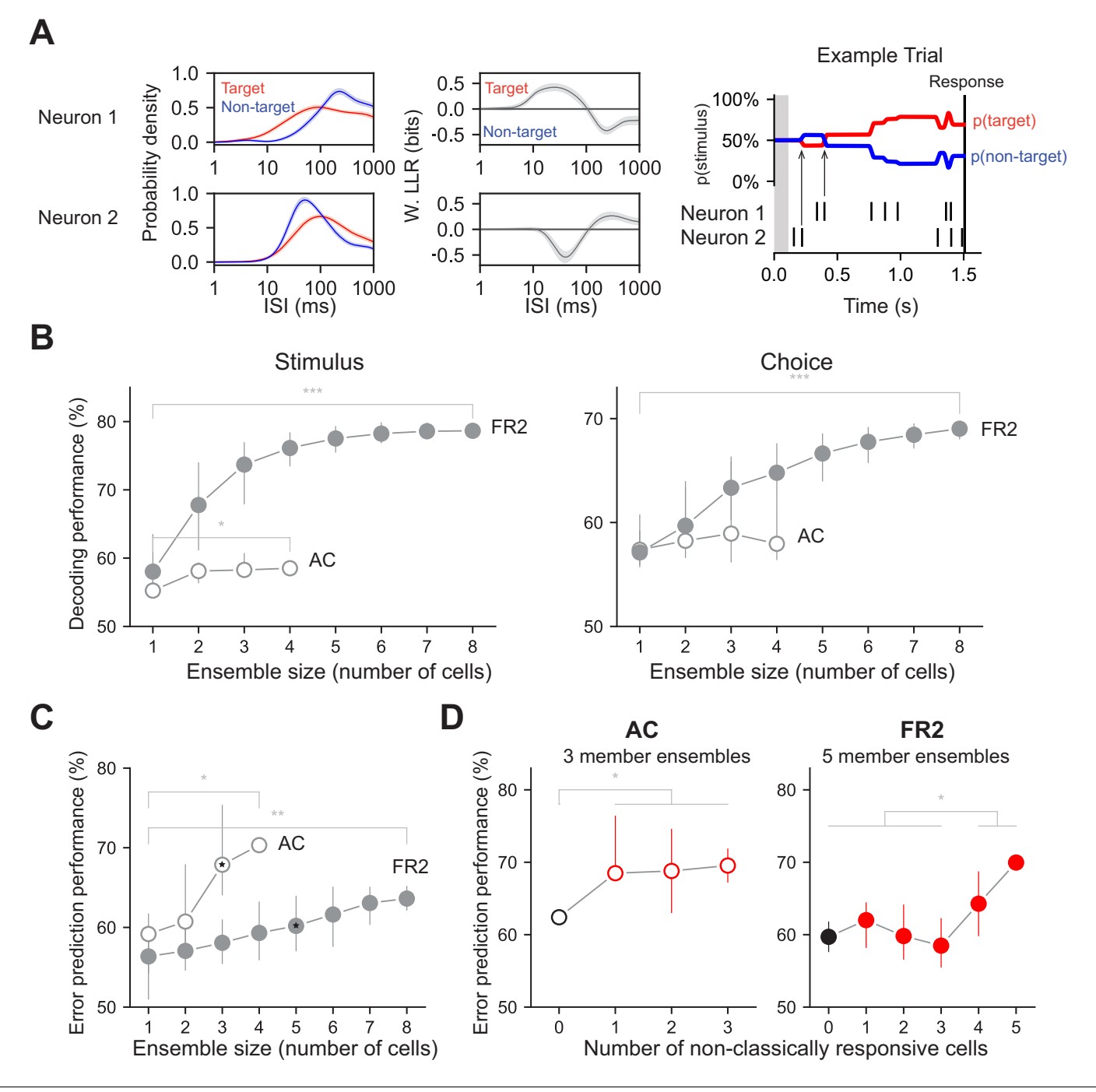

**Figure 7.** Decoding performance of neuronal ensembles recorded in AC or FR2. (**A**) Schematic of ensemble decoding. Left, conditional ISI distributions and corresponding weighted LLR shown for two simultaneously recorded neurons. Right, an example trial where each neuron's ISIs and LLRs are used to independently update stimulus category according to Bayes' rule. Arrows indicate the first updates from each neuron. (**B**) Stimulus and choice decoding performance for ensembles in AC and FR2 for ensembles of increasing size (Comparing smallest with largest ensembles. Stimulus: $*p_{AC} = 0.04$, $***p_{FR2} = 1 \times 10^{-5}$, Choice: $p_{AC} = 0.29$, $***p_{FR2} = 7 \times 10^{-5}$, Mann-Whitney U test, two-sided). (**C**) Error prediction performance in AC and FR2 as a function of ensemble size ($*p_{AC} = 0.03$, $**p_{FR2} = 0.002$; comparison between AC and FR2, for 3-member ensembles: $p=1.2 \times 10^{-5}$, for 4-member ensembles: $p=0.03$, Mann-Whitney U test, two-sided). Chance performance is 50%. (**D**) Error prediction performance in AC and FR2 as a function of the number of non-classically responsive cells in the ensemble ($*p_{AC} = 0.037$, Welch's t-test with Bonferroni correction for multiple comparisons; $*p_{FR2} = 0.015$, Student's t-test with Bonferroni correction), 3 and 5 member ensembles in c. shown for AC and FR2 respectively. Chance performance is 50%.

DOI: https://doi.org/10.7554/eLife.42409.023

*Figure 7 continued on next page*

*Figure 7 continued*

The following figure supplement is available for figure 7:

**Figure supplement 1.** PSTHs from two example cells recorded in either AC or FR2 separated by correct (top) and error (bottom) trials.
DOI: https://doi.org/10.7554/eLife.42409.024

trials (*Figure 8E* stimulus-aligned, dotted line, Δconsensus, $t$ = 0 to 0.81 s, $p_{SR}$ = 0.14 Wilcoxon test with Bonferroni correction, two-sided). For choice-related activity, choice non-classically responsive ensembles in both regions as well as choice classically responsive ensembles in FR2 each reached consensus within 500 ms of the behavioral response (*Figure 8E*, response-aligned, Δconsensus, $t$ = −1.0 to 0.0 s, $p_{CNR}$ = 2.0 × $10^{-5}$, $p_{CR}$ = 0.12 Wilcoxon test with Bonferroni correction, two-sided). Importantly, this temporally precise pattern of consensus building is not present on error trials. On error trials, stimulus consensus dynamics decreased over the course of the trial whereas choice dynamics did not display a systematic increase with the exception of choice non-classically responsive ensembles in AC which remained systematically lower than correct trials (*Figure 8F*, Δconsensus, correct trials vs. error trials, stimulus: $p_{SNR}$ = 0.007, $p_{SR}$ = 0.065, choice: $p_{CNR}$ = 0.0048, $p_{CR}$ = 0.065 Mann-Whitney U test, two-sided, Δconsensus on error trials, $t$ = 0 to 0.42 s, $p_{SNR}$ = 1.3 × $10^{-33}$, $t$ = −1.0 to 0 s, $p_{CNR}$ = 0.032, $p_{CR}$ = 0.14 Wilcoxon test with Bonferroni correction, two-sided). The observed increases in ensemble consensus on correct trials (while failing do so on error trials) suggests that achieving a shared ISI representation of task variables may be relevant for successful task execution.

These results reveal that consensus-building and divergence occur at key moments during the trial for successful execution of behavior in a manner that is invisible at the level of the PSTH. As sensory and choice non-classically responsive ensembles participated in these dynamics, changes in the consensus value cannot simply be a byproduct of correlated firing rate modulation due to tone-evoked responses or ramping. While consensus-building can only indicate a shared representation, divergence can indicate one of two things: (1) the LLRs of each cell within an ensemble are completely dissimilar or (2) they are 'out of phase' with one another – the LLRs partition the ISIs the same way (*Figure 8D*, dotted lines), but the same ISIs code for opposite behavioral variables. This distinction is important because (2) implies coordinated structure of ensemble activity (the partitions of the ISI align) whereas (1) does not. To distinguish between these two possibilities, we used the 'unsigned consensus', a second measure sensitive to the ISI partitions but insensitive to the sign of the LLR. Both 'in phase' and perfectly 'out of phase' LLRs would produce an unsigned consensus of 1, whereas unrelated LLRs would be closer to 0 (*Figure 8D*). For example, in the second row of *Figure 8D*, both cells agree that ISIs < 100 ms indicate one stimulus category and ISIs > 100 ms indicate another, but they disagree about which set of ISIs mean target and which mean non-target. This results in a consensus value of 0 (out of phase) but an unsigned consensus value of 1.

Using this metric, we found that the unsigned consensus pattern for non-classically responsive ensembles (ensembles with two or more non-classically responsive members) were shared between AC and FR2 – increasing until ~750 ms after tone onset on correct trials (*Figure 8G*, stimulus-aligned, Δ consensus, $t$ = 0 to 0.89 s, p=1.7 × $10^{-5}$ Wilcoxon test, two-sided). Non-classically responsive ensembles in AC and FR2 also increased their unsigned consensus immediately before behavioral response (although values in AC were lower overall; *Figure 8G*, response-aligned, Δconsensus, $t$ = −1.0 to 0.0 s, p=0.0011 Wilcoxon test, two-sided). This pattern of consensus-building was only present on correct trials. On error trials unsigned consensus values did not systematically increase (*Figure 8H*, Δconsensus compared to error trials, p=1.9 × $10^{-9}$ Mann-Whitney U test, two-sided) suggesting that behavioral errors might result from a general lack of consensus between ensemble members. In summary, we have shown that cells which appear unmodulated during behavior do not encode task information independently, but do so by synchronizing their representation of behavioral variables dynamically during the trial.

## Discussion

Using a straightforward, single-trial, ISI decoding algorithm that makes few assumptions about the proper model for neural activity, we found task-specific information extensively represented by non-classically responsive neurons in both AC and FR2 that lacked conventional task-related, trial-

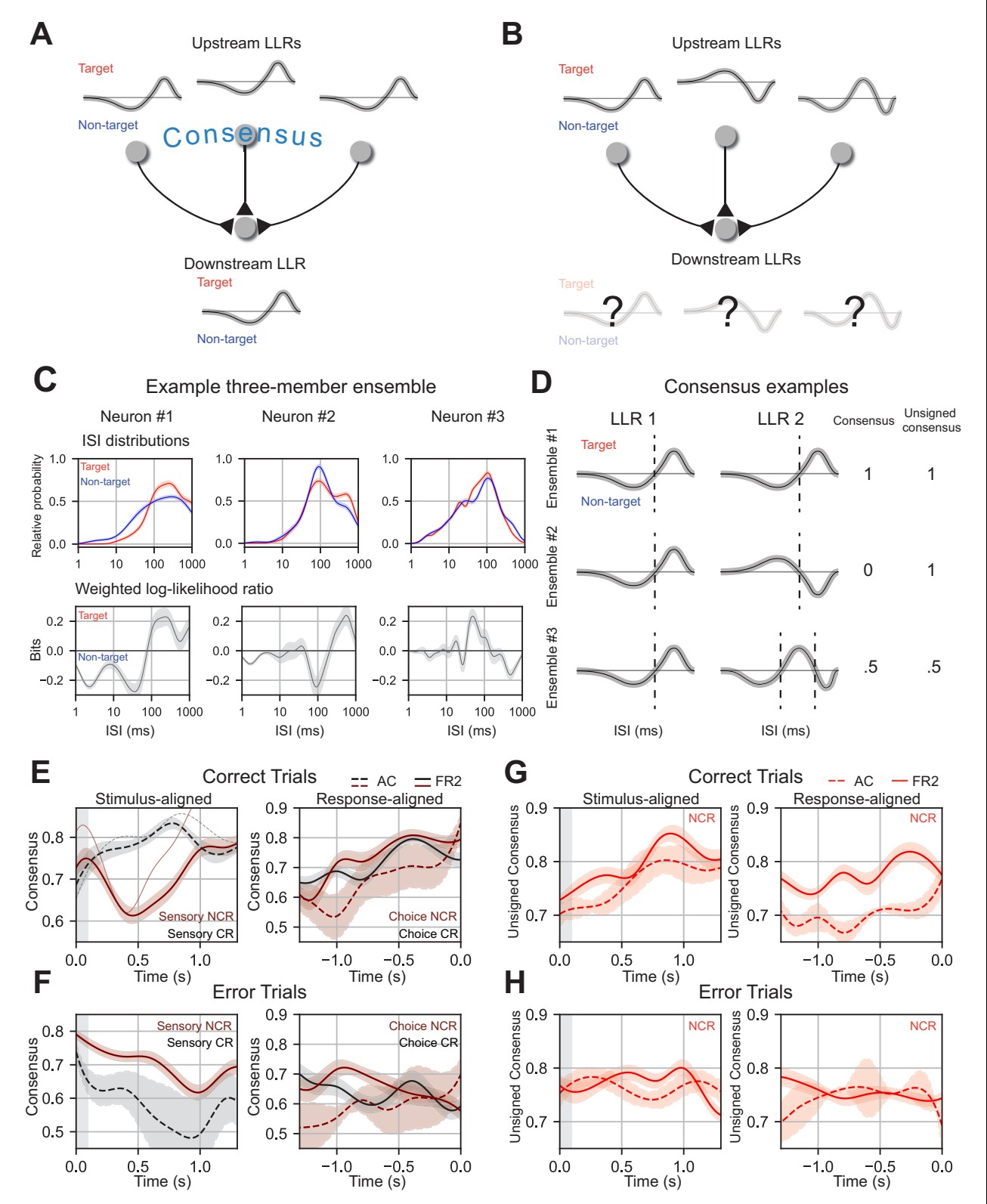

**Figure 8.** . Ensemble consensus-building during behavior. (**A**) Schematic of consensus building in a three-member ensemble. When the LLRs of ensemble members are similar the meaning of any ISI is unambiguous to a downstream neuron. (**B**) Schematic of a three-member ensemble without consensus. The meaning of an ISI depends on the upstream neuron it originates from. (**C**) ISI distributions, and LLRs for three members of a sample ensemble. Note that despite differences in ISI distributions, neuron #1 and neuron #2 have similar weighted log-likelihood ratios (ISIs > 200 ms indicate

*Figure 8 continued on next page*

*Figure 8 continued*

target, ISIs < 200 ms indicate non-target). (D) Consensus values for three illustrative two-member ensembles. Ensemble 1 members have identical LLRs, agreeing on the meaning of all ISIs (consensus = 1) and on how the ISIs should be partitioned (unsigned consensus = 1). Ensemble 2 contains cells with LLRs where the ISI meanings are reversed, disagreeing on meaning of the ISIs (consensus = 0) but still agree on how the ISIs should be partitioned (unsigned consensus = 1). Ensemble 3 contains two cells with moderate agreement about the ISI meanings and partitioning, leading to intermediate consensus and unsigned consensus values (0.5 for each). (E) Left, mean consensus as a function of time from tone onset (stimulus-aligned) on correct trials for three-member sensory classically responsive ensembles in AC (two or more members sensory classically responsive; black dotted line; n=11 ensembles) and sensory non-classically responsive ensembles in FR2 (two or more members sensory non-classically responsive; dark red solid line; n=101 ensembles). Standard deviation shown around each mean trendline. Thin solid and dotted line represent an individual consensus trajectory from FR2 and AC, respectively. FR2 sensory non-classically responsive cells consistently reached consensus and then diverged immediately after stimulus presentation ($\Delta$consensus, $t = 0$ to $0.42$ s, $p_{SNR} = 3.9 \times 10^{-4}$ Wilcoxon test with Bonferroni correction, two-sided). AC classically responsive ensembles (black) increase consensus until 750 ms ($\Delta$consensus, $t = 0$ to $0.81$ s, $p_{SR} = 0.14$ Wilcoxon test with Bonferroni correction, two-sided). Right, mean consensus as a function of time to behavioral response (response-aligned) on correct trials for three-member choice classically responsive ensembles (two or more members choice classically responsive; black) in FR2 (solid line; n=47 ensembles) and choice non-classically responsive (two or more members choice non-classically responsive; dark red) in AC (dotted line; n=11 ensembles) and FR2 (solid line; n=57 ensembles). Standard deviation shown around each mean trendline. On correct trials, choice classically responsive (black) and choice non-classically responsive ensembles (dark red) in both regions reached high consensus values ~500 ms before response ($\Delta$consensus, $t = -1.0$ to $0.0$ s, $p_{CNR} = 2.0 \times 10^{-5}$, $p_{CR} = 0.12$ Wilcoxon test with Bonferroni correction, two-sided). (F) As in e, but for error trials ($\Delta$consensus, correct vs. error trials, stimulus: $p_{SNR} = 0.007$, $p_{SR} = 0.065$, choice: $p_{CNR} = 0.0048$, $p_{CR} = 0.065$ Mann-Whitney U test, two-sided). (G) Unsigned consensus index for non-classically responsive ensembles (two or more members non-classically responsive) in AC (dotted line; n=13 ensembles) and FR2 (solid line; n=36 ensembles), stimulus-aligned (left, $\Delta$consensus, $t = 0$ to $0.89$ s, $p = 5.1 \times 10^{-5}$ Wilcoxon test **with Bonferroni correction**, two-sided) and response-aligned (right, $\Delta$consensus, $t = -1.0$ to $0.0$ s, $p = 0.0033$ Wilcoxon test with Bonferroni correction, two-sided). On correct trials, ensembles reach high values of unsigned consensus ~750 ms after tone onset and within 500 ms of behavioral response. (H) As in (G), but for error trials ($\Delta$consensus, correct vs. error trials, $p = 1.9 \times 10^{-9}$ Mann-Whitney U test, two-sided). (E) – (G) Combinations analyzed and shown are those for which there are significant numbers in our dataset.

DOI: https://doi.org/10.7554/eLife.42409.025

averaged firing rate modulation. The complexity of single-trial spiking patterns and the apparent variability between trials led to the development of this novel decoding method. Furthermore, the heterogeneity in the observed ISI distributions within and across brain regions precluded a straightforward interpretation of these distributions and instead suggested an approach which focused on whether and when these distributions are shared in local ensembles via consensus-building.

The degree to which single neurons were task-modulated was uncorrelated with conventional response properties including frequency tuning. AC and FR2 each represent both task-variables; furthermore, in both regions, we identified many multiplexed neurons that simultaneously represented the sensory input and the upcoming behavioral choice including non-classically responsive cells. This highlights that the cortical circuits that generate behavior exist in a distributed network – blurring the traditional modular view of sensory and frontal cortical regions.

Most notably, FR2 has a better representation of task-relevant auditory stimuli than AC. The prevalence of stimulus information in FR2 might be surprising given that AC reliably responds to pure tones in untrained animals; however, when tones take on behavioral significance, this information is encoded more robustly in frontal cortex, suggesting that this region is critical for identifying the appropriate sensory-motor association. Furthermore, the stark improvement in stimulus encoding for small ensembles in FR2 suggests that task-relevant stimulus information is reflected more homogeneously in local firing activity across FR2 (perhaps through large-scale ensemble consensus-building) while this information is reflected in a more complex and distributed manner throughout AC.

We have identified task-informative non-classically responsive neurons recorded while animals performed a frequency recognition task or a task-switching paradigm. This does not preclude the possibility that these cells are driven by other acoustic stimuli or in other behavioral contexts; however, determining the significance of non-classically responsive activity must ultimately be considered in the specific behavioral context in question, as their role may be dynamic and context dependent.

The finding that the ISI-based approach of our algorithm is not reducible to rate despite their close mathematical relationship raises the question of how downstream regions could respond preferentially to specific ISIs. Our whole-cell recordings from both AC and FR2 demonstrate that different postsynaptic cells can respond differently to the same input pattern with a fixed overall rate, emphasizing the importance of considering a code sensitive to precise spike-timing perhaps via

mechanisms of differential short-term plasticity such as depression and facilitation (*Figure 5—figure supplement 1*). Furthermore, this is supported by experimental and theoretical work showing that single neurons can act as resonators tuned to a certain periodicity of firing input (*Izhikevich, 2000*). This view could also be expanded to larger neuronal populations comprised of feedback loops that would resonate in response to particular ISIs. In this case, cholinergic neuromodulation could offer a mechanism for adjusting the sensitivities of such a network during behavior on short time-scales by providing rapid phasic signals (*Hangya et al., 2015*).

Our consensus results reveal dynamic changes in the relationship between the LLRs of ensemble members. How might such a downstream resonator interpret a given ISI in the context of these dynamics? Our consensus analysis provides one possible answer: downstream neurons may be attuned to the ISIs specified by the consensus LLR of an ensemble. In such a model, an ensemble would have the strongest influence on downstream activity when they reach high consensus. We additionally hypothesize that mechanisms of long-term synaptic plasticity such as spike-timing-dependent plasticity can redistribute synaptic efficacy, essentially changing the dynamics of short-term plasticity independent from overall changes in amplitudes (*Markram and Tsodyks, 1996*). Thus, after training, downstream neurons do not need to continually change the readout mechanism-rather, the upstream and downstream components might be modified together by cortical plasticity during initial phases of behavioral training. This would set the ISI distributions appropriate for firing of task-relevant downstream neurons, which would ensure that ensemble consensus is reached for correct sensory processing in highly trained animals.

It is still unclear what the relevant timescales of decoding might be in relation to phenomena such as membrane time constants, periods of oscillatory activity, and behavioral timescales. Given that our ISI-based decoder and conventional rate-modulated decoders reveal distinct information, future approaches might hybridize these rate-based and temporal-based decoding methods to span multiple timescales. Other recent studies have also contributed to our understanding of non-classically responsive activity, by evaluating firing rates or responses from calcium imaging to demonstrate how correlations with classically responsive activity may contribute to the linear separability of ensemble responses (*Leavitt et al., 2017*; *Zylberberg, 2018*).

We have shown that underlying the task-relevant information encoded by each ensemble is a rich set of consensus-building dynamics that is invisible at the level of the PSTH. Ensembles in both FR2 and AC underwent stimulus and choice-related consensus building that was only observed when the animal correctly executed the task. Moreover, non-classically responsive cells demonstrated temporal dynamics synchronized across regions which were distinct from classically responsive ensembles. These results underscore the importance of measuring neural activity in behaving animals and using unbiased and generally applicable analytical methods, as the response properties of cortical neurons in a behavioral context become complex in ways that challenge our conventional assumptions (*Carcea et al., 2017*; *Fritz et al., 2010*; *Kuchibhotla et al., 2017*; *Otazu et al., 2009*).

## Materials and methods

### Key resources table

| Reagent type (species) or resource | Designation | Source or reference | Identifiers | Additional information |
|---|---|---|---|---|
| Strain, strain background (*Rattus norvegicus domesticus*, males and females) | Sprague-Dawley, rats | Charles River, Taconic | NTac:SD | |
| Chemical compound, drug | Muscimol | Sigma-Aldrich | InChi:ZJQHPWUVQPJPQT -UHFFFAOYSA-N; SID:24896662 | |
| Software, algorithm | Single-trial Bayesian decoding algorithm | newly created | N/A | https://github.com/ badralbanna /Insanally2017 |
| Other | *Rodgers and DeWeese, 2014* dataset | CRCNS | pfc-1 | http://crcns.org/data-sets/pfc/pfc-1 |

## Behavior

All animal procedures were performed in accordance with National Institutes of Health standards and were conducted under a protocol approved by the New York University School of Medicine Institutional Animal Care and Use Committee. We used 23 adult Sprague-Dawley male and female rats (Charles River) in the behavioral studies. Animals were food restricted and kept at 85% of their initial body weight, and maintained at a 12 hr light/12 hr dark cycle.

Animals were trained on a go/no-go audiomotor task (*Carcea et al., 2017*; *Froemke et al., 2013*). Operant conditioning was performed within 12' L x 10' W x 10.5' H test chambers with stainless steel floors and clear polycarbonate walls (Med Associates), enclosed in a sound attenuation cubicle and lined with soundproofing acoustic foam (Med Associates). The nose and reward ports were both arranged on one of the walls with the speaker on the opposite wall. The nose port, reward port, and the speaker were controlled and monitored with a custom-programmed microcontroller. Nose port entries were detected with an infrared beam break detector. Auditory stimuli were delivered through an electromagnetic dynamic speaker (Med Associates) calibrated using a pressure field microphone (ACO Pacific).

Animals were rewarded with food for nose poking within 2.5 s of presentation of the target tone (4 kHz) and given a short 7 s time-out for incorrectly responding to non-target tones (0.5, 1, 2, 8, 16, 32 kHz). Incorrect responses include either failure to enter the nose port after target tone presentation (miss trials) or entering the nose port after non-target tone presentation (false alarms). Tones were 100 ms in duration and sound intensity was set to 70 dB SPL. Tones were presented randomly with equal probability such that each stimulus category was presented. The inter-trial interval delays used were 5, 6, 7, or 8 s.

For experiments involving muscimol, we implanted bilateral cannulas in either FR2 (+2.0 to +4.0 mm AP, ±1.3 mm ML from Bregma) of seven animals or AC (-5.0 to -5.8 mm AP, 6.5-7.0 mm ML from Bregma) of three animals. We infused 1 µL of muscimol per side into FR2 or infused 2 µL of muscimol per side into AC, at a concentration of 1 mg/mL. For saline controls, equivalent volumes of saline were infused in each region. Behavioral testing was performed 30-60 min after infusions. Power analysis was performed to determine sample size for statistical significance with a power of β: 0.8; these studies required at least three animals, satisfied in the experiments of Figure 1-figure supplement 3B,E. For motor control study, animals could freely nose poke for food reward without presentation of auditory stimuli after muscimol and saline infusion.

## Implant preparation and surgery

Animals were implanted with microdrive arrays (Versadrive-8 Neuralynx) in either AC (eight animals) or FR2 (seven animals) after reaching behavioral criteria of $d' \geq 1.0$. For surgery, animals were anesthetized with ketamine (40 mg/kg) and dexmedetomidine (0.125 mg/kg). Stainless steel screws and dental cement were used to secure the microdrive to the skull, and one screw was used as ground. Each drive consisted of eight independently adjustable tetrodes. The tetrodes were made by twisting and fusing four polyimide-coated nichrome wires (Sandvik Kanthal HP Reid Precision Fine Tetrode Wire; wire diameter 12.5 µm). The tip of each tetrode was gold-plated to an impedance of 300–400 kOhms at 1 kHz (NanoZ, Neuralynx).

## Electrophysiological recordings and unit isolation

Recordings in behaving rats were performed as previously described (*Carcea et al., 2017*). After the animal recovered from surgery (~7 days) recordings began once performance returned to pre-surgery levels. Tetrodes were advanced ~60 µm 12 hr prior to each recording session, to a maximum of 2.5 mm (for FR2) or 2.0 mm (for AC) from the pial surface. For recording, signals were first amplified onboard using a small 16-bit unity-gain preamplifier array (CerePlex M, Blackrock Microsystems) before reaching the acquisition system. Spikes were sampled at 30 kS/sec and bandpass filtered between 250 Hz and 5 kHz. Data were digitized and all above-threshold events with signal-to-noise ratios > 3:1 were stored for offline spike sorting. Single-units were identified on each tetrode using OfflineSorter (Plexon Inc) by manually classifying spikes projected as points in 2D or 3D feature space. The parameters used for sorting included the waveforms projection onto the first two principal components, energy, and nonlinear energy. Artifacts were rejected based on refractory period violations (<1 msec). Clustering quality was assessed based on the Isolation Distance and $L_{ratio}$

sorting quality metrics. To be initially included for analysis, cells had to have >3 spikes per trial for 80% of trials to ensure that there were enough ISIs to reliably estimate the ISI probability density functions.

## Statistical tests for non-classical responsiveness

We used two positive statistical tests for non-classical responsiveness: one to establish a lack of tone-modulation, the other to establish a lack of ramping activity. To accommodate the possibility of tone onset and offset responses, we performed our tone-modulation test on a 100 ms long tone presentation window as well as the 100 ms window immediately after tone presentation. The test compared the number of spikes during each of these windows to inter-trial baseline activity as measured by three sequential 100 ms windows preceding tone onset. Three windows were chosen to account for variability in spontaneous spike counts. Given that spike counts are discrete, bounded, and non-normal, we used subsampled bootstrapping to evaluate whether the mean change in spikes during tone presentation was sufficiently close to zero (in our case 0.1 spikes). We subsampled 90% of the spike count changes from baseline, calculated the mean of these values, and repeated this process 5000 times to construct a distribution of means. If 95% of the subsampled means values were between −0.1 and 0.1, we considered the cell sensory non-classically responsive (p<0.05). The range of mean values from −0.1 to 0.1 were included to account for both tone-evoked (increases in spike count) and tone-suppressed (decreases in spike count) activity. The value of 0.1 spikes was chosen to be conservative as it is equivalent to an expected change of 1 spike every 10 trials. This is a conservative, rigorous method for establishing sensory non-classical responsiveness that is commensurate with more standard approaches for establishing tone responsiveness such as the z-score.

To quantify the observed sustained increase or decrease in firing rate preceding the behavioral response, a ramp index was calculated adapted from the 'build-up rate' used in previous literature[31]. First, the trial averaged firing rate was determined in 50 ms bins leading up to the behavioral response. We then calculated the slope of a linear regression in a 500 ms long sliding window beginning 850 ms before behavioral response. The maximum value of these slopes was used as the 'ramp index' for each cell. Cells were classified as choice non-classically responsive if the ramp index did not indicate an appreciable change in the firing rate (less than 50% change) established via subsampled bootstrapping. Cells that were shown to be both sensory and choice non-classically responsive were considered non-classically responsive overall (*Figure 4A,B*, red circles).

## Additional firing statistics

Spontaneous average firing rate was established by averaging spikes in a 100 ms time window immediately prior to tone onset on each trial. To quantify tone modulated responses observed during stimulus presentation, we calculated z-scores of changes in spike count from 100 ms before tone onset to 100 ms during tone presentation:

$$z = \frac{\mu}{\sigma}$$

where $\mu$ is the mean change in spike count and $\sigma$ is the standard deviation of the change in spike count.

## Analysis of receptive field properties

Receptive fields were constructed by calculating the average change in firing rate from 50 ms before tone onset to 50 ms during tone presentation. The window used during tone presentation was identical to that used to calculate the z-score. Best frequency was defined as the frequency where the largest positive deviation in the evoked firing rate was observed. Tuning curve bandwidth was determined by calculating the width of the tuning curve measured at the mean of the maximum and minimum observed evoked firing rates.

## In vivo whole-cell recordings

Sprague-Dawley rats 3–5 months old were anesthetized with pentobarbital. Experiments were carried out in a sound-attenuating chamber. Series of pure tones (70 dB SPL, 0.5–32 kHz, 50 ms, 3 ms cosine on/off ramps, inter-tone intervals between 50 and 500 ms) were delivered in pseudo-random

sequence. Primary AC location was determined by mapping multiunit responses 500–700 µm below the surface using tungsten electrodes. In vivo whole-cell voltage-clamp recordings were then obtained from neurons located 400–1100 µm below the pial surface. Recordings were made with an AxoClamp 2B (Molecular Devices). Whole-cell pipettes (5–9 MΩ) contained (in mM): 125 Cs-gluconate, 5 TEACl, 4 MgATP, 0.3 GTP, 10 phosphocreatine, 10 HEPES, 0.5 EGTA, 3.5 QX-314, 2 CsCl, pH 7.2. Data were filtered at 2 kHz, digitized at 10 kHz, and analyzed with Clampfit 10 (Molecular Devices). Tone-evoked excitatory postsynaptic currents were recorded at –70 mV.

## In vitro whole-cell recordings

Acute brain slices of AC or FR2 were prepared from 2 to 5 month old Sprague-Dawley rats. Animals were deeply anesthetized with a 1:1 ketamine/xylazine cocktail and decapitated. The brain was rapidly placed in ice-cold dissection buffer containing (in mM): 87 NaCl, 75 sucrose, 2.5 KCl, 1.25 NaH2PO4, 0.5 CaCl2, 7 MgCl2, 25 NaHCO3, 1.3 ascorbic acid, and 10 dextrose, bubbled with 95%/5% O2/CO2 (pH 7.4). Slices (300–400 µm thick) were prepared with a vibratome (Leica), placed in warm dissection buffer (32–35°C) for 10 min, then transferred to a holding chamber containing artificial cerebrospinal fluid at room temperature (ACSF, in mM: 124 NaCl, 2.5 KCl, 1.5 MgSO4, 1.25 NaH2PO4, 2.5 CaCl2, and 26 NaHCO3,). Slices were kept at room temperature (22–24°C) for at least 30 min before use. For experiments, slices were transferred to the recording chamber and perfused (2–2.5 ml min$^{-1}$) with oxygenated ACSF at 33°C. Somatic whole-cell current-clamp recordings were made from layer five pyramidal cells with a Multiclamp 700B amplifier (Molecular Devices) using IR-DIC video microscopy (Olympus). Patch pipettes (3–8 MΩ) were filled with intracellular solution containing (in mM): 120 K-gluconate, 5 NaCl, 10 HEPES, 5 MgATP, 10 phosphocreatine, and 0.3 GTP. Data were filtered at 2 kHz, digitized at 10 kHz, and analyzed with Clampfit 10 (Molecular Devices). Focal extracellular stimulation was applied with a bipolar glass electrode (AMPI Master-9, stimulation strengths of 0.1–10 V for 0.3 msec). Spike trains recorded from AC and FR2 units during behavior were then divided into 150–1000 msec fragments, and used as extracellular input patterns for these recordings.

## ISI-based single-trial Bayesian decoding

Our decoding method was motivated by the following general principles: First, single-trial spike timing is one of the only variables available to downstream neurons. Any observations about trial-averaged activity must ultimately be useful for single-trial decoding, in order to have behavioral significance. Second, there may not be obvious structure in the trial-averaged activity to suggest how non-classically responsive cells participate in behaviorally-important computations. This consideration distinguishes our method from other approaches that rely explicitly or implicitly on the PSTH for interpretation or decoding (*Churchland et al., 2008*; *Erlich et al., 2011*; *Jaramillo et al., 2014*; *Jaramillo and Zador, 2011*; *Murakami et al., 2014*; *Wiener and Richmond, 2003*). Third, we required a unified approach capable of decoding from both classically responsive and non-classically responsive cells in sensory and frontal areas with potentially different response profiles. Fourth, our model should contain as few parameters as possible to account for all relevant behavioral variables (stimulus category and behavioral choice). This model-free approach also distinguishes our method from others that rely on parametric models of neural activity.

These requirements motivated our use of ISIs to characterize neuronal activity. For non-classically responsive cells with PSTHs that displayed no systematic changes over trials or between task conditions, the ISI distributions can be variable. The ISI defines spike timing relative to the previous spike and thus does not require reference to an external task variable such as tone onset or behavioral response. In modeling the distribution of ISIs, we use a non-parametric Kernel Density Estimator that avoids assumptions about whether or not firing occurs according to a Poisson (or another) parameterized distribution. We used 10-fold cross validation to estimate the bandwidth of the Gaussian kernel in a data-driven manner. Finally, the use of the ISI was also motivated by previous work demonstrating that the ISI can encode sensory information (*Lundstrom and Fairhall, 2006*; *Reich et al., 2000*; *Zuo et al., 2015*) and that precise spike timing has been shown to be important for sensory processing in rat auditory cortex (*DeWeese et al., 2003*; *Lu and Wang, 2004*). Our data-driven method combines (1) non-parametric statistical procedures (Kernel Density Estimation), (2) use of the ISI as the response variable of interest (rather than an estimate of the instantaneous

firing rate locked to an external task variable), and (3) single-trial decoding via Bayesian inference rendering it a novel decoder capable of decoding responsive as well as non-classically responsive activity from any brain region.

## Training probabilistic model

Individual trials were defined as the time from stimulus onset to the response time of the animal (or average response time in the case of no-go trials). Trials were divided into four categories corresponding to each of the four possible variable combinations (target/go, target/no-go, non-target/go, non-target/no-go). Approximately 90% of each category was set aside as a training set in order to determine the statistical relationship between the ISI and the two task variables (stimulus category, behavioral choice).

Each ISI observed was sorted into libraries according to the stimulus category and behavioral choice of the trial. The continuous probability distribution of finding a particular ISI given the task condition of interest (target or non-target, go or no-go) was then inferred using nonparametric Kernel Density Estimation with a Gaussian kernel of bandwidth set using a 10-fold cross-validation (*Jones et al., 1996*). Because the domain of the distribution of ISIs is by definition positive (ISI > 0), the logarithm of the ISI was used to transform the domain to all real numbers. In the end, we produced four continuous probability distributions quantifying the probability of observing an ISI on a trial of a given type: p(ISI|target), p(ISI|non-target), p(ISI|go), and p(ISI|no-go). These distributions were estimated in a 1 s long sliding window (recalculated every 100 ms) starting at the beginning of the trial and ending at the end of the trial to account for dynamic changes in the ISI distributions over the course of the trial. These likelihood functions assume that the observed ISIs are independent of the previous spiking history of the cell. While this assumption is violated in practice, estimation of the joint probability of an ISI and previous ISIs using non-parametric methods was infeasible given to the limited number of ISI combinations observed over the session without including additional assumptions about the correlation structure between ISIs.

## Decoding

The remaining 10% of trials in the test set are then decoded using the ISI likelihood function described in the previous section. Each trial begins with agnostic beliefs about the stimulus category and the upcoming behavioral choice (p(target)=p(non-target)=50%). Each time an ISI was observed, beliefs were updated according to Bayes' rule with the four probability distributions obtained in the previous section serving as the likelihood function. To update beliefs in the probability of the target tone when a particular ISI has been observed, we used the following relationship:

$$p(\text{target}|\text{ISI, t}) = \frac{p(\text{ISI}|\text{target, t})p(\text{target, t})}{p(\text{ISI}|\text{target, t})p(\text{target, t}) + p(\text{ISI}|\text{non}-\text{target,t})p(\text{non}-\text{target,t})}$$

On the left hand side are the updated beliefs about the probability of a target. When the next ISI is observed this value would be inserted as p(target, t) on the right side of the equation and updated once more. Using the probability normalization, p(non-target, t) can be determined,

$$p(\text{target, t}) + p(\text{non}-\text{target,t}) = 1$$

Similarly, for choice,

$$p(\text{go}|\text{ISI, t}) = \frac{p(\text{ISI}|\text{go, t})p(\text{go, t})}{p(\text{ISI}|\text{go, t})p(\text{go, t}) + p(\text{ISI}|\text{no}-\text{go,t})p(\text{no}-\text{go,t})}$$

and

$$p(\text{go, t}) + p(\text{no}-\text{go,t}) = 1$$

As the likelihood functions were estimated in 1 s long sliding windows recalculated every 100 ms, Each ISI was assessed using the likelihood function that placed the final spike closest to the center of the sliding window.

Continuing this process over the course of the trial, we obtain four probabilities – one for each of the variable outcomes – as a function of time during the trial: p(target, t), p(non-target, t), p(go, t),

and p(no-go, t). At each moment, the total probability of both stimuli and both choices are 1. The prediction for the entire trial was assessed at the end of the trial, using the overall likelihood function. Given our independence assumption, the overall likelihood for a spike train is simply equal to product of the likelihoods for each ISI observed over the course of the trial,

$$p(\{ISI_i\} \,|\, target) = \prod_{i=1}^{n} p(ISI_i \,|\, target, t_i).$$

We used 10-fold cross-validation, meaning the trials in the four stimulus categories were randomly divided into ten parts and each part took a turn acting as the test set with the remaining 90% of trials acting as a training set. To estimate the statistical certainty of these results we used bootstrapping with 124 repetitions (except in the case of the null hypotheses where 1240 repetitions were used).

### Ensemble decoding

Ensemble decoding proceeded very similarly to the single-unit case. The ISI probability distributions for each neuron in the ensemble were calculated independently as described above. However, while decoding a given trial, the spike trains of all neurons in the ensemble were used to simultaneously update the beliefs about stimulus category and behavioral choice. In other words, p(stimulus, t) and p(choice, t) were shared for the entire ensemble but each neuron updated them independently using Bayes' rule whenever a new ISI was encountered. Correlations between neurons were ignored and each of the ISIs from each cell were assumed to were assumed to be independent. For example, if an ISI is observed at time $t$ from neuron $j$ with a likelihood $p_j$:

$$p(target|ISI, t) = \frac{p_j(ISI|target, t)p(target, t)}{p_j(ISI|target, t)p(target, t) + p_j(ISI|non-target, t)p(non-target, t)}$$

This process is repeated every time a new ISI is encountered from any cell in the ensemble.

The joint likelihood of observing a set of ISIs during a trial is then the product of the likelihoods of each neuron independently. For example, for a two neuron ensemble, the combined likelihood, $p_{12}$, of observing the set $\{ISI_i\}_1$ from neuron 1 and $\{ISI_i\}_2$ from neuron 2 is:

$$p_{12}\big(\{ISI_i\}_1, \{ISI_i\}_2 \big| target\big) = p_1\big(\{ISI_i\}_1 \,\big|\, target\big)p_2\big(\{ISI_i\}_2 \,\big|\, target\big)$$

where $p_j$ is the likelihood of observing a given set of ISIs from neuron $j$.

## Synthetic spike trains

To test the null hypothesis that the ISI-based single-trial Bayesian decoder performance was indistinguishable from chance, synthetic spike trains were constructed for each trial of a given unit by randomly sampling with replacement from the set of all observed ISIs regardless of the original task variable values (synthetic spike trains, *Figure 4E*). In principle under this condition, ISIs should no longer bear any relationship to the task variables and decoding performance should be close to 50%. For single-unit responses, this randomization was completed 1240 times. Significance from the null was assessed by a direct comparison to the 124 bootstrapped values observed from the true data to the 1240 values observed under the null hypotheses. The p-value was determined as the probability of finding a value from this synthetic condition that produced better decoding performance than the values actually observed as in a standard permutation test.

As a secondary control, we used a traditional permutation test whereby observed spike trains were left intact, but the task variables that correspond to each spike train were randomly permuted (condition permutation, *Figure 4F*). This process was completed 1240 times.

## Rate-modulated Poisson decoding

To decode using the trial-averaged firing rate, we implemented a standard method (*Rieke et al., 1999*) which uses the probability of observing a set of $n$ spikes at times $t_1, \ldots, t_n$ assuming those spikes were generated by a rate-modulated Poisson process (*Figure 4—figure supplement 4*). Just as with this ISI-based decoder, we decoded activity from the entire trial. First, we use a training set comprising 90% of trials to estimate the time-varying firing rate for each condition from the PSTH

$(r_{\text{target}}(t),\ r_{\text{non-target}}(t),\ r_{\text{go}}(t), r_{\text{no-go}}(t))$ by Kernel Density Estimation with 10-fold cross-validation. The remaining 10% of spike trains are then decoded using the probability of observing each spike train on each condition assuming they were generated according to a rate-modulated Poisson process:

$$p(\{t_i\}\,|\text{target}) = \frac{1}{N!}\left(r_{\text{target}}(t_1)\,r_{\text{target}}(t_2)\dots r_{\text{target}}(t_n)\right)\exp\left(-\int_{T_i}^{T_f} r_{\text{target}}(t)\,dt\right),$$

where $T_i$ and $T_f$ are the beginning and end of the trial respectively. This likelihood function is straightforward to interpret: the first product is the probability of observing spikes the spikes at the times they were observed (where the 1/*N!* term serves to divide out by the number of permutations of spike labels) and the exponential term represents the probability of silence in the periods between spikes. For comparison with our method, we can reformulate this equation using interspike intervals, if we first break up the exponential integral into domains that span the observed interspike intervals.

$$p(\{t_i\}|\text{target})$$
$$= \frac{1}{N!}\left(r_{\text{target}}(t_1)\exp\left(-\int_{T_i}^{t_1} r_{\text{target}}(t)\,dt\right)\right)$$
$$\times\left(r_{\text{target}}(t_2)\exp\left(-\int_{t_1}^{t_2} r_{\text{target}}(t)\,dt\right)\right)\dots\times\left(\exp\left(-\int_{t_n}^{T_f} r_{\text{target}}(t)\,dt\right)\right).$$

Collecting the first and last terms relating to trial start and trial end as

$$L_i(t_1,\ T_i)\equiv r_{\text{target}}(t_1)\exp\left(-\int_{T_i}^{t_1} r_{\text{target}}(t)\,dt\right)$$

$$L_f(t_n, T_f)\equiv\exp\left(-\int_{t_n}^{T_f} r_{\text{target}}(t)\,dt\right),$$

this becomes

$$p(\{t_i\}\,|\text{target}) = \frac{1}{N!}L_i\left(\prod_{i=1}^{n-1} r_{\text{target}}(t_i+\Delta t_i)\,\exp\left(-\int_{t_i}^{t_i+\Delta t_i} r_{\text{target}}(t)\,dt\right)\right)L_f,$$

where $\Delta t_i$ is the time difference between spikes $t_i$ and $t_{i+1}$. The interpretation of each term in the product is straightforward: it is the infinitesimal probability of observing a spike a time $\Delta t$ after a spike at time $t$ multiplied by the probability of observing no spikes in the intervening time. In other words, it is simply $p(\text{ISI}\,|\,\text{target},t)$, the probability of observing an ISI conditioned on observing the first spike at time *t*, as predicted by the assumption of a rate-modulated Poisson process. We can easily verify that this term is normalized which allows us to write,

$$p(\text{ISI}\,|\,\text{target},t) = r_{\text{target}}(t+\text{ISI})\,\exp\left(-\int_t^{t+\text{ISI}} r_{\text{target}}(t)\,dt\right).$$

With the exception of the terms relating to trial start and end, we can then view the likelihood of a spike train as resulting from the likelihood of the individual ISIs (just as with our ISI-decoder),

$$p(\{t_i\}\,|\text{target}) = \frac{1}{N!}L_i\,L_f\left(\prod_{i=1}^{n-1} p(\text{ISI}_i\,|\,\text{target},\ t_i)\right),$$

with the key difference that these ISI probabilities are inferred from the firing rate rather than estimated directly using non-parametric methods.

## Inferring the ISI distribution predicted by a rate-modulated Poisson process

To compare the ISI distribution inferred using non-parametric methods to one predicted by a rate-modulated Poisson process, we use the relationship above to calculate the predicted probability of observing an ISI of given length within the 1 s window used for our non-parametric estimates. The formula above assumes a spike has already occurred at time *t*, so we multiply by the probability of

observing a spike at time $t$, $\mathrm{p}(t \mid \mathrm{target}) = r_{\mathrm{target}}(t)$, to obtain the total probability of finding an ISI at any given point in the trial.

$$\begin{aligned} \mathrm{p}(\mathrm{ISI}, t \mid \mathrm{target}) &= \mathrm{p}(\mathrm{ISI} \mid \mathrm{target}, \mathrm{t})\mathrm{p}(\mathrm{t}|\mathrm{target}) \\ &= r_{\mathrm{target}}(t)\, r_{\mathrm{target}}(t + \mathrm{ISI})\, \exp\left(-\int_t^{t+\mathrm{ISI}} r_{\mathrm{target}}(t)\, dt\right). \end{aligned}$$

In other words, the probability of observing an ISI beginning at time t is simply the probability of observing spikes at times $t$ and $t$ + ISI with silence in between.

The probability of observing an ISI at *any* time within a time window spanning $w_i$ to $w_f$ is simply the integral of this ISI probability as a function of time across the window. To ensure the final spike occurs before $w_f$ the integral spans $w_i$ to ($w_f$ - ISI),

$$p(\mathrm{ISI} \mid w_I, w_f, \mathrm{target}) = C^{-1} \int_{w_i}^{w_f - \mathrm{ISI}} p(\mathrm{ISI}, t \mid \mathrm{target}) dt$$

where $C$ is a normalization constant which ensures p(ISI | $w_i$, $w_f$, target) integrates to 1,

$$C = \int_0^{w_f - w_i} \left(\int_{w_i}^{w_f - \mathrm{ISI}} p(\mathrm{ISI}, t \mid \mathrm{target}) dt\right) d\mathrm{ISI}.$$

## Regression-based method for verifying multiplexing

For each cell, we fit a Logit model for both the stimulus and choice decoding probabilities on individual trials with the true stimulus category and behavioral choice as regressors. We then calculated the extent to which the stimulus decoding probability was determined by true stimulus category by subtracting the regression coefficient for stimulus from that of choice (*Figure 4—figure supplement 3A,x*-axis, stimulus selectivity index); when this number is positive it indicates that stimulus was a stronger predictor of stimulus decoding on a trial-by-trial basis. The same process was repeated for choice (*Figure 4—figure supplement 3A,y*-axis, choice selectivity index). According to this analysis, we took multiplexed cells to be those that were positive for both measures (*Figure 4—figure supplement 3A*, orange symbols, 19/90 cells). In other words, multiplexed cells were cells for which stimulus decoding probabilities were primarily a result of true stimulus category *and* choice decoding probabilities were primarily a result of true behavioral choice.

Given the moderate negative correlation for these indices, we projected each of these points onto their linear regression to create a one-dimensional regression-based uniplexing index. Cells with a value near zero are the multiplexed cells described above and cells with positive or negative values are primarily stimulus or choice selective (*Figure 4—figure supplement 3A*).

We compared the uniplexing values produced by this regression method to those produced by examining only the average decoding performance for stimulus and choice (*Figure 4—figure supplement 3B*). A decoding-based uniplexing index was defined as the difference between average stimulus and choice decoding for each cell. When these two values are comparable, this measure returns a value close to zero and the cell is considered multiplexed; moreover, cells that are uniplexed for stimulus or choice receive positive and negative values, respectively, just as with the regression-based measure. While the overall magnitude of these two measures need not be related, both measures of multi/uniplexing rank cells on a one-dimensional axis from choice uniplexed to multiplexed to stimulus uniplexed centered on zero.

## Weighted log likelihood ratio

The log likelihood ratio (LLR) was calculated by first calculating the conditional ISI probabilities and then taking the difference of the logarithm of these distributions. For stimulus,

$$\mathrm{LLR}_{\mathrm{stimulus}}(\mathrm{ISI}) = \log_2(\mathrm{p}(\mathrm{ISI}|\mathrm{target})) - \log_2(\mathrm{p}(\mathrm{ISI}|\mathrm{non-target})),$$

and for choice,

$$\mathrm{LLR}_{\mathrm{choice}}(\mathrm{ISI}) = \log_2(\mathrm{p}(\mathrm{ISI}|\mathrm{go})) - \log_2(\mathrm{p}(\mathrm{ISI}|\mathrm{no-go})).$$

The weighted LLR weights the LLR according to the prevalence of a given ISI. For stimulus,

$$\mathrm{W.LLR_{stimulus}(ISI)} = \mathrm{p(ISI)}[\log_2(\mathrm{p(ISI|target)}) - \log_2(\mathrm{p(ISI|non-target)})],$$

and for choice,

$$\mathrm{W.LLR_{choice}(ISI)} = \mathrm{p(ISI)}[\log_2(\mathrm{p(ISI|go)}) - \log_2(\mathrm{p(ISI|no-go)})].$$

## Consensus and unsigned consensus

The consensus value evaluates the extent to which the LLR (or weighted LLR) is shared across an ensemble. It is the norm of the sum of the LLRs (W. LLRs) divided by the sum of the norms. In principle, the functional norm can be anything but in this case we used the *l*1 norm (the absolute area under the curve),

$$\|f\|_1 \equiv \int |f(x)|\,dx.$$

The for an n-member ensemble, the consensus is then

$$\mathrm{Consensus} \equiv \frac{\left\|\sum_{i=1}^{n}\mathrm{LLR}_i\right\|_1}{\sum_{i=1}^{n}\|\mathrm{LLR}_i\|_1}.$$

For the *unsigned consensus*, we first generate every permutation of the LLRs used and their inverses, -LLR, up to an overall sign. For example, for a pair of LLRs there are only two options,

$$(\mathrm{LLR}_1, \mathrm{LLR}_2)\,\mathrm{or}\,(\mathrm{LLR}_1, -\mathrm{LLR}_2),$$

and for three LLRs there are four options,

$$(\mathrm{LLR}_1, \mathrm{LLR}_2, \mathrm{LLR}_3), (-\mathrm{LLR}_1, \mathrm{LLR}_2, \mathrm{LLR}_3), (\mathrm{LLR}_1, -\mathrm{LLR}_2, \mathrm{LLR}3), \mathrm{or}\,(\mathrm{LLR}_1, \mathrm{LLR}_2, -\mathrm{LLR}_3)$$

The consensus is then calculated over each these sets and the maximum value is taken to be the value of the unsigned consensus.

To generate the consensus curves in *Figure 8*, LLRs are calculated using a 750 ms sliding window recalculated every 100 ms. The resulting consensus value is assigned to the center of the 750 ms window. For visual clarity, these values were interpolated by a third-degree univariate spline calculated using the python package *scipy.interpolate.InterpolatedUnivariateSpline* (this technique is guaranteed to intercept the measured values).

## Analysis of *Rodgers and DeWeese (2014)* dataset

Using our novel ISI-based decoding algorithm, we analyzed cells found to be non-classically responsive in a previously published study (*Rodgers and DeWeese, 2014*). Briefly, rats were trained on a novel auditory stimulus selection task where animals had to respond to one of two cues while ignoring the other depending on the context. Rats held their nose in a center port for 250 to 350 ms and were then presented with two simultaneous sounds (a white noise burst played from only the left or right speaker and a high- or low-pitched warble played from both speakers). In the 'localization' context, animals were trained to ignore the warble and respond to the location of the white noise burst and in the 'pitch' context they were trained to ignore the location of the white noise burst and respond to the pitch of the warble. Cells recorded from both primary auditory cortex and prefrontal cortex (prelimbic region) were shown to be classically responsive to the selection rule during the pre-stimulus period (i.e. firing rates differed between the two contexts). Non-classically responsive cells were reported but not further analyzed.

We established that cells were non-classically responsive for the stimulus location or pitch using our own positive statistical criteria for non-classical responsiveness (described above) by comparing the average spiking activity in the 250 ms stimulus period and the 250 ms following stimulus to inter-trial baseline activity. Cells were also determined to be non-classically responsive for ramping using the same criteria as with our own data. We confirmed that cells were non-classically responsive for the selection rule by comparing their average spiking activity in the 100 ms immediately preceding stimulus onset across contexts.

To determine whether non-classically responsive cells also encoded task information (stimulus location, stimulus pitch, behavioral choice, and the selection rule), we decoded each variable on single-trials using our ISI-based decoding algorithm. Selection rule information was only assessed in the pre-stimulus hold period, whereas stimulus and choice information was assessed in the period after stimulus onset prior to behavioral response (as with our own data). Cells shown in *Figure 5B* were deemed statistically significant when compared to the decoding performance of a control using synthetically generated data (p<0.05).

## Statistical analysis

All statistical analyses were performed in Python, MATLAB, or GraphPad Prism 6. Datasets were tested for normality, and appropriate statistical tests applied as described in the text (e.g. Student's paired t-test for normally distributed data, Mann-Whitney U test for unmatched non-parametric data, and Wilcoxon matched-pairs signed rank test for matched non-parametric data).

## Code and sample data availability

https://github.com/badralbanna/Insanally2017 (*Albanna, 2019*; copy archived at https://github.com/elifesciences-publications/Insanally2017).

## Acknowledgements

We thank E Simoncelli, ND Daw, CS Peskin, AA Fenton, E Kelemen, K Kuchibhotla, E Morina, M Aoi, and A Charles for comments, discussions, and technical assistance, and CA Loomis and the NYU School of Medicine Histology Core for assistance with anatomical studies. Shari E Ross produced the illustration in *Figure 1A*. This work was funded by an NYU Provost's Postdoctoral Fellowship and NIDCD (DC015543-01A1) to MNI; a NARSAD Young Investigators Award, NIMH (T32), and NIMH (KMH106744A) to IC; a James McDonnell Understanding Human Cognition Scholar Award to KR; a Fordham University Grant for Cloud Based Computing Research Projects to BFA; and NIDCD (DC009635 and DC012557), the NYU Grand Challenge Award, a Howard Hughes Medical Institute Faculty Scholarship, a Hirschl/Weill-Caulier Career Award, and a Sloan Research Fellowship to RCF The authors declare no competing financial interests.

## Additional information

### Funding

| Funder | Grant reference number | Author |
| --- | --- | --- |
| National Institute on Deafness and Other Communication Disorders | DC015543 | Michele N Insanally |
| New York University | Provost's Postdoctoral Fellowship | Michele N Insanally |
| National Alliance for Research on Schizophrenia and Depression | Young Investigators Award | Ioana Carcea |
| National Institute of Mental Health | T32 | Ioana Carcea |
| National Institute of Mental Health | KMH106744A | Ioana Carcea |
| James S. McDonnell Foundation | Understanding Human Cognition Scholar Award | Kanaka Rajan |
| Fordham University | Grant for Cloud Based Computing Research Projects | Badr F Albanna |
| Howard Hughes Medical Institute | Faculty Scholarship | Robert C Froemke |
| Alfred P. Sloan Foundation | Sloan Research Fellowship | Robert C Froemke |

| | | |
|---|---|---|
| National Institute on Deafness and Other Communication Disorders | DC009635 | Robert C Froemke |
| National Institute on Deafness and Other Communication Disorders | DC012557 | Robert C Froemke |
| New York University | Grand Challenge Award | Robert C Froemke |
| Albert Einstein College of Medicine | Hirschl/Weill-Caulier Career Award, | Robert C Froemke |

The funders had no role in study design, data collection and interpretation, or the decision to submit the work for publication.

## Author contributions

Michele N Insanally, Conceptualization, Resources, Data curation, Software, Formal analysis, Funding acquisition, Validation, Investigation, Visualization, Methodology, Writing—original draft, Project administration, Writing—review and editing; Ioana Carcea, Funding acquisition, Investigation, Methodology, Writing—review and editing; Rachel E Field, Chris C Rodgers, Investigation; Brian DePasquale, Kanaka Rajan, Software, Formal analysis, Methodology, Writing—review and editing; Michael R DeWeese, Conceptualization, Software, Supervision, Methodology, Writing—review and editing; Badr F Albanna, Conceptualization, Resources, Data curation, Software, Formal analysis, Funding acquisition, Validation, Investigation, Methodology, Writing—review and editing; Robert C Froemke, Conceptualization, Supervision, Funding acquisition, Project administration, Writing—review and editing

## Author ORCIDs

Brian DePasquale http://orcid.org/0000-0002-3830-3184
Badr F Albanna http://orcid.org/0000-0002-5536-6443
Robert C Froemke http://orcid.org/0000-0002-1230-6811

## Ethics

Animal experimentation: All animal procedures were performed in accordance with National Institutes of Health standards and were conducted under a protocol (#160611-03) approved by the New York University School of Medicine Institutional Animal Care and Use Committee.

## Decision letter and Author response

Decision letter https://doi.org/10.7554/eLife.42409.028
Author response https://doi.org/10.7554/eLife.42409.029

# Additional files

## Supplementary files

• Transparent reporting form
DOI: https://doi.org/10.7554/eLife.42409.026

## Data availability

The code and data underlying our analyses are freely available online (https://github.com/badralbanna/Insanally2017; copy archived at https://github.com/elifesciences-publications/Insanally2017).

The following datasets were generated:

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
