## [Decision Letter]

Thank you for submitting your article "Nominally non-responsive frontal and sensory cortical cells encode task-relevant variables via ensemble consensus" for consideration by *eLife*. Your article has been reviewed by three peer reviewers, and the evaluation has been overseen by a Reviewing Editor and Timothy Behrens as the Senior Editor. The reviewers have opted to remain anonymous.

The reviewers found the work interesting, making a substantive and impactful contribution to the field. They have discussed the reviews with one another and the Reviewing Editor has drafted this decision to help you prepare a revised submission.

Summary:

Insanally et al. ask how information about auditory stimuli and behavioral choices, are reflected within the spiking patterns of neurons in the auditory cortex (AC), and frontal cortical area 2 (FR2) of rat. The subjects were presented with octave spaced tones, and required to select go/no-go to the presence of a target tone. The work focuses on neurons whose firing rates (PSTH) show no change in response to the auditory stimulus, and/or no ramping activity preceding the decision. The main finding is that timing information in the ISIs of these neurons can be decoded to determine the stimulus and/or choice. The work demonstrates this using a nonparametric decoding method. Specifically, the authors develop a novel method for quantifying the difference in ISI distributions corresponding to different task-conditions. They use this method to provide a posterior probability of task-condition given the ISIs in a single trial. The median decoding performance is moderate but statistically significant, and superior to rate/PSTH-based metrics.

Essential revisions:

1) The paper refers to neurons with no stimulus-dependent firing rate modulation as "nominally non-responsive", however, these neurons clearly do respond to the stimulus (by changing their ISI distributions). Indeed, this is the basis for the decoding analysis. Consequently, it seems like the main result is that the spike timing contains information that is missed if one only looks for stimulus dependence in the firing rates. This is an important point and worth making, but is lost by labeling these neurons as non-responsive. The paper would be much clearer if the authors would be more upfront in the title, Abstract, Results, etc., about what their data show: timing information is there even when the PSTH looks uninformative.

2) The neural network model in Figure 7 lacks some relevance to the data. A rate-based model is used, in which some neurons have stimulus-locked rate modulations, and some don't, and show that inactivating these "non-responsive" neurons changes task performance (presumably due to the effects of those inactivations on the rest of the network through the recurrent connectivity). In the experimental data, however, the authors show that, even when the neurons' rates don't have obvious stimulus dependence, the timing information still does, and any effects of inactivating subsets of neurons are left untested. To make the model relevant, it would need to use spiking neuron models, and recapitulate the basic phenomenon from the experiments: spike timing information is present even when there's no rate information. Alternatively, the authors could do inactivation experiments (e.g., with archaerhodopsin) to test the main prediction of the current model, which would make it relevant. This experiment would probably be hard to do, it is suggested instead that the authors revise or remove the computational model.

3) Relatedly, the conjecture in the last paragraph of the subsection “Ensemble consensus-building dynamics underlie hidden task information”, and again somewhat in the last paragraph of the Discussion, that the NR cells are a separate functional network from the responsive ones contradicts the modeling result. If they are separate networks, silencing the NR cells wouldn't affect the responsive ones. To retain the claim about separate functional networks, it seems necessary to do optogenetics, or some other manipulation of activity, and show that changing NR cells' activity doesn't affect responsive ones. Otherwise, the authors should remove this unsupported claim.

4) The decoder is quite nice, but it makes several independence assumptions that are not borne out in real neural populations. This point should be made much more upfront in the Materials and methods and the Results: stating that this decoder approximates Bayesian inference, by ignoring all forms of correlation. Doing this will ensure that readers don't mistakenly think these simple expressions are exact. Alternatively, the decoder could be updated to include the correlations discussed below.

Correlations: in the subsection “Decoding” the first, third and sixth equations, and the equation in the subsection “Ensemble decoding”, assume that each ISI is independent for a given neuron, which is not generally true, due to AHP currents among other things. The paper also assumes that ISIs are independent between neurons, which is generally not true in simultaneous neural recordings, due to noise correlations, etc.

5) Two other recent works, listed below, showed a stronger form of the claim advanced in the title and Abstract: that neurons with no stimulus dependence can still contribute to the neural code. There, the non-responsive neurons individually contain no stimulus information, although through correlations with other cells, they still contribute to population-wide neural coding.

The current paper's results are distinct from those, since they consider timing information, whereas the other papers consider rates. However, it would be worth making that distinction somewhere in the paper.

a) Leavitt et al., 2017.

b) Zylberberg, 2017.

6) From the Materials and methods (subsection “Training probabilistic model”), it looks like the ISI distributions used for the decoder are recalculated for different time points during the integration period. This is not clear at all from the text in the Results section, but is a crucial point, as it puts severe limits on how a downstream neural circuit could read out this information: the downstream circuit would need to continually change its readout "decoder" somehow. The decoder and analysis still have a lot of merit. They show that the stimulus information is present in the timing, and show how it could be extracted (e.g., for BMI applications). But the time dependence needs to be made clear in the Results section, and the biological plausibility or implausibility should be discussed.

7) Please expand the Materials and methods and Results sections to clarify the following points:

a) Temporal resolution. It seems like the ISI distributions are estimated in a 1s window (subsection “Training probabilistic model”). Sliding by what amount? How is it decided which 1s windows each ISI contributes? Only those that contain the 2 spikes enclosing the ISI? Are predictions are evaluated at the end of each of these 1s windows (so the first prediction is made at second 1 after trial onset)?

b) Clarify where the formula for the likelihood of the full spike train (subsection “Decoding”, last paragraph) is used. From reading the manuscript, it seems that the formulas for updating the posterior only consider likelihoods for a given ISI.

c) Ensemble decoding. Please be more explicit and include a mathematical formula for how the updating of the posterior is done in this case.

d) Subsection “Ensemble consensus-building dynamics underlie hidden task information”, second paragraph. Sliding window for consensus. Is this the same 1s sliding window above? Probably not, given the time-varying nature of the curves and the fact that they start at zero. Please add explanatory text about the time windows used in the consensus section in the methods. Please also describe in more detail how the curves in Figure 8E-H were calculated?

e) Furthermore, is it correct that the Poisson process estimator takes into account the full duration of the stimulus-response interval? There is some asymmetry in the information available with the ISI approach (which takes into account the 1+ seconds following stimulus onset) and the PSTH (which is short, given the 100ms tone). Were there any firing rate differences over the longer interval that would predict responses?

8) Please add a careful discussion of whether the animal's choice can be cleanly dissected from stimulus encoding, particularly given the very high performance of these animals on the task, the long duration between stimulus and response, and the late build-up of stimulus prediction/consensus. There is plenty of literature on the effects of attention, behavior, and working memory on cortical synchronization and timing, albeit mostly in hippocampus. It is possible that the AC responses simply reflect the choice/memory already decided. The manuscript attempts to disambiguate the choice from the stimulus with a log linear regression, but this is dependent upon prediction information from the decoder analysis that itself may already contain mixed information.

9) Title: Please consider revising your title to either omit or make more intuitive what "ensemble consensus" means and to clarify "nominally non-responsive", given the concerns of some of the reviewers about this terminology. One suggested revision "Response timing in frontal and sensory cortical neurons encodes task-relevant variables even when average responses are uninformative".

---

## [Author Response]

Essential revisions:1) The paper refers to neurons with no stimulus-dependent firing rate modulation as "nominally non-responsive", however, these neurons clearly do respond to the stimulus (by changing their ISI distributions). Indeed, this is the basis for the decoding analysis. Consequently, it seems like the main result is that the spike timing contains information that is missed if one only looks for stimulus dependence in the firing rates. This is an important point and worth making, but is lost by labeling these neurons as non-responsive. The paper would be much clearer if the authors would be more upfront in the title, Abstract, Results, etc., about what their data show: timing information is there even when the PSTH looks uninformative.

To avoid possible confusion, we have taken the reviewer’s suggestion and we have modified the manuscript (including the title, Abstract, main text, and figures) to better describe these neurons. We have changed all references to “nominally non-responsive” neurons to “non-classically responsive” neurons. Our title now reads: “Spike-timing-dependent ensemble encoding by non-classically responsive cortical neurons.” We have also modified the text to emphasize that spike timing reveals task-related information in non-classically responsive neurons even when the PSTH does not (see Abstract; main text; Discussion, first paragraph).

2) The neural network model in Figure 7 lacks some relevance to the data. A rate-based model is used, in which some neurons have stimulus-locked rate modulations, and some don't, and show that inactivating these "non-responsive" neurons changes task performance (presumably due to the effects of those inactivations on the rest of the network through the recurrent connectivity). In the experimental data, however, the authors show that, even when the neurons' rates don't have obvious stimulus dependence, the timing information still does, and any effects of inactivating subsets of neurons are left untested. To make the model relevant, it would need to use spiking neuron models, and recapitulate the basic phenomenon from the experiments: spike timing information is present even when there's no rate information. Alternatively, the authors could do inactivation experiments (e.g., with archaerhodopsin) to test the main prediction of the current model, which would make it relevant. This experiment would probably be hard to do, it is suggested instead that the authors revise or remove the computational model.

We agree with the reviewer and, following their suggestion, we have removed the neural network model from this manuscript. The suggestion to make the model more relevant by using spiking neurons has inspired us to instead include it in a separate, follow-up study where we can more extensively investigate network dynamics.

3) Relatedly, the conjecture in the last paragraph of the subsection “Ensemble consensus-building dynamics underlie hidden task information”, and again somewhat in the last paragraph of the Discussion, that the NR cells are a separate functional network from the responsive ones contradicts the modeling result. If they are separate networks, silencing the NR cells wouldn't affect the responsive ones. To retain the claim about separate functional networks, it seems necessary to do optogenetics, or some other manipulation of activity, and show that changing NR cells' activity doesn't affect responsive ones. Otherwise, the authors should remove this unsupported claim.

We agree that this point requires further clarification and substantiation and have removed this claim. To clarify, we did not intend the term “separate” to indicate that nominally non-responsive ensembles are independent from responsive ensembles, only that they may play a differential role during behavior (a claim supported by their ability to predict behavioral errors, Figure 7, and distinct consensus dynamics, Figure 8).

4) The decoder is quite nice, but it makes several independence assumptions that are not borne out in real neural populations. This point should be made much more upfront in the Materials and methods and the Results: stating that this decoder approximates Bayesian inference, by ignoring all forms of correlation. Doing this will ensure that readers don't mistakenly think these simple expressions are exact. Alternatively, the decoder could be updated to include the correlations discussed below.Correlations: in the subsection “Decoding” the first, third and sixth equations, and the equation in the subsection “Ensemble decoding”, assume that each ISI is independent for a given neuron, which is not generally true, due to AHP currents among other things. The paper also assumes that ISIs are independent between neurons, which is generally not true in simultaneous neural recordings, due to noise correlations, etc.

We have added text to the Results section and Materials and methods (subsection “Training probabilistic model”) to clarify this point. The Results section now reads, “These ISI likelihood functions consider each ISI to be independent of previous ISIs and therefore ignore correlations between ISIs”. The decision to exclude correlations was made on practical grounds: we had initially attempted to include correlations between an ISI and the preceding ISI in the likelihood function, but in order to properly estimate this conditional likelihood function using non-parametric methods (in sparsely firing cortical neurons) would require a much larger number of trials than what our dataset allows. The same issue applies when estimating the joint probability of ISIs between neurons. In earlier implementations of our ensemble decoder, we instead tried including relative spike timing between cells (i.e., the interspike interval between a spike in cell 1 and one in cell 2) but this factor did not result in significantly better decoding performance overall.

5) Two other recent works, listed below, showed a stronger form of the claim advanced in the title and Abstract: that neurons with no stimulus dependence can still contribute to the neural code. There, the non-responsive neurons individually contain no stimulus information, although through correlations with other cells, they still contribute to population-wide neural coding.The current paper's results are distinct from those, since they consider timing information, whereas the other papers consider rates. However, it would be worth making that distinction somewhere in the paper.a) Leavitt et al., 2017.b) Zylberberg, 2018.

We appreciate the suggestion to put our work in context of these two studies; we now cite both of these papers. In the Discussion section, we have added text on the complementarity of their results with our own which reads, “Other recent studies have also contributed to our understanding of non-classically responsive activity, by evaluating firing rates or responses from calcium imaging to demonstrate how correlations with classically responsive activity may contribute to the linear separability of ensemble responses (Leavitt et al., 2017; Zylberberg, 2018).”

6) From the Materials and methods (subsection “Training probabilistic model”), it looks like the ISI distributions used for the decoder are recalculated for different time points during the integration period. This is not clear at all from the text in the Results section, but is a crucial point, as it puts severe limits on how a downstream neural circuit could read out this information: the downstream circuit would need to continually change its readout "decoder" somehow. The decoder and analysis still have a lot of merit. They show that the stimulus information is present in the timing, and show how it could be extracted (e.g., for BMI applications). But the time dependence needs to be made clear in the Results section, and the biological plausibility or implausibility should be discussed.

We thank the reviewer for pointing this out, as this is an important detail that should be made clear. We included a moving window to take into account non-stationarity in the ISI distributions over the course of the trial and interestingly this decision did improve decoding performance in practice. The observed dynamics of the ISI distributions both in our decoding results and our consensus building results raise the question of how a downstream cell would interpret the ISIs of these cells. Assuming this downstream cell was attuned to properly interpret one set of ISI distributions, which set should it choose? Our consensus results suggest an answer: it may be that downstream cells interpret upstream activity in terms of the consensus LLRs – that way when ensemble members disagree on a representation their activity is unlikely to drive activity in the downstream cell, but when their representation is aligned with each other and the downstream cell they can reliably influence downstream activity.

We hypothesize that long-term synaptic plasticity can adapt spike timings during the initial phases of behavioral training, leading to stable representations and ISI distributions across cortical networks. One of the major predictions of our algorithm is that forms of short-term synaptic plasticity (e.g., paired-pulse depression or facilitation) are important aspects of downstream decoding of complex spike trains. This is supported by past theoretical and experimental work (Abbott et al., 1997). Furthermore, mechanisms of long-term synaptic plasticity such as spike-timing-dependent plasticity can redistribute synaptic efficacy, essentially changing the dynamics of short-term plasticity independent from overall changes in amplitudes (Markram and Tsodyks, 1996). Thus we wouldn’t necessarily expect that the downstream circuit needs to continually change the readout mechanism – rather, the upstream and downstream components might be modified together over the course of behavioral training, to set the ISI distributions appropriate for firing of task-relevant downstream neurons, which would ensure that ensemble consensus is reached for correct sensory processing in highly-trained animals.

We have clarified the time dependence of the ISI distributions in the Results section by adding text that reads, “To accommodate any non-stationarity, these ISI distributions were calculated in 1 second long sliding windows over the course of the trial.” We have also added text regarding the biological plausibility in the Discussion section. It now reads: “Our whole-cell recordings from both AC and FR2 demonstrate that different postsynaptic cells can respond differently to the same input pattern with a fixed overall rate, emphasizing the importance of considering a code sensitive to precise spike-timing perhaps via mechanisms of differential short-term plasticity such as depression and facilitation”, and “Our consensus results reveal dynamic changes in the relationship between the LLRs of ensemble members. […] This would set the ISI distributions appropriate for firing of task-relevant downstream neurons, which would ensure that ensemble consensus is reached for correct sensory processing in highly-trained animals.”

7) Please expand the Materials and methods and Results sections to clarify the following points:a) Temporal resolution. It seems like the ISI distributions are estimated in a 1s window (subsection “Training probabilistic model”). Sliding by what amount? How is it decided which 1s windows each ISI contributes? Only those that contain the 2 spikes enclosing the ISI? Are predictions are evaluated at the end of each of these 1s windows (so the first prediction is made at second 1 after trial onset)?

The algorithm calculated the ISI distributions in 1 second windows recalculated every 100 ms. Every time an ISI was encountered, the algorithm consulted the distribution generated in a window as close to centered around the final spike as possible and evaluated the prediction update at the time of the final spike. As mentioned above to point #6, we have clarified this point in the Results section, “To accommodate any non-stationarity, these ISI distributions were calculated in 1 second long sliding windows recalculated every 100 ms over the course of the trial.” And in the Materials and methods section “These distributions were estimated in a 1 second long sliding window (recalculated every 100 ms).”

b) Clarify where the formula for the likelihood of the full spike train (subsection “Decoding”, last paragraph) is used. From reading the manuscript, it seems that the formulas for updating the posterior only consider likelihoods for a given ISI.

Our method updates sequentially every time an ISI is encountered. The formula for the overall likelihood function was included for conceptual clarity: to calculate the final prediction for the entire trial (as opposed to dynamics of the prediction during the trial shown in Figure 3D) using the likelihood function for the entire trial is sufficient. We have added text to the Materials and methods to clarify this point: “The prediction for the entire trial was assessed at the end of the trial, using the overall likelihood function.”

c) Ensemble decoding. Please be more explicit and include a mathematical formula for how the updating of the posterior is done in this case.

We have clarified this by adding text and including a mathematical formula in the Materials and methods section, "For example, if an ISI is observed at time *t* from neuron *j* with a likelihood p*_j_*:ptarget|ISI,t=pjISItarget,tp(target,t)pjISItarget,tptarget,t+pjISInon-target, tp(non-target, t)

This process is repeated every time a new ISI is encountered from any cell in the ensemble.”.

d) Subsection “Ensemble consensus-building dynamics underlie hidden task information”, second paragraph. Sliding window for consensus. Is this the same 1s sliding window above? Probably not, given the time-varying nature of the curves and the fact that they start at zero. Please add explanatory text about the time windows used in the consensus section in the methods. Please also describe in more detail how the curves in Figure 8E-H were calculated?

Yes, the intuition is correct, the sliding window used was not the same as for the above. To achieve improved temporal precision, we used a 750 ms sliding window to calculate the LLRs and the consensus values every 100 ms. The consensus values calculated were assigned to the middle of the window and the series of values were interpolated with a third-degree univariate spline for visual clarity (this method generates smooth curves that are guaranteed to intercept the calculated values). Figure 8E-H represent the means and standard deviations of these curves for the relevant region and ensemble type. We have clarified these points in the Materials and methods section, “To generate the consensus curves in Figure 8,LLRs are calculated using a 750 ms sliding window recalculated every 100 ms. […] For visual clarity, these values were interpolated by a third-degree univariate spline calculated using the python package *scipy.interpolate.InterpolatedUnivariateSpline* (this technique is guaranteed to intercept the measured values).”

e) Furthermore, is it correct that the Poisson process estimator takes into account the full duration of the stimulus-response interval? There is some asymmetry in the information available with the ISI approach (which takes into account the 1+ seconds following stimulus onset) and the PSTH (which is short, given the 100ms tone). Were there any firing rate differences over the longer interval that would predict responses?

Yes, both the Poisson process estimator and the ISI-based decoder took into account the full trial activity (from stimulus onset to behavioral response). We clarified this in the Materials and methods: “Just as with this ISI-based decoder, we decoded activity from the entire trial.”

8) Please add a careful discussion of whether the animal's choice can be cleanly dissected from stimulus encoding, particularly given the very high performance of these animals on the task, the long duration between stimulus and response, and the late build-up of stimulus prediction/consensus. There is plenty of literature on the effects of attention, behavior, and working memory on cortical synchronization and timing, albeit mostly in hippocampus. It is possible that the AC responses simply reflect the choice/memory already decided. The manuscript attempts to disambiguate the choice from the stimulus with a log linear regression, but this is dependent upon prediction information from the decoder analysis that itself may already contain mixed information.

We thank the reviewer for the opportunity to address this point. For the reasons the reviewer mentions, a clean dissection of stimulus and choice variables is a challenging endeavor. This is less of a concern for uniplexed neurons which only contain information about one of the two variables, which is why our multiple regression analysis focused on establishing that the multiplexed neurons were not simply a trivial byproduct of correlations between the behavioral variables. The *assumption* of this approach was that the decoding results may contain mixed information, and the objective was to evaluate the extent to which a decoder’s predictions (e.g., the stimulus decoder) were correlated with not only the relevant behavioral variable (e.g., stimulus) but also the other variable (e.g., choice). While the results from this analysis demonstrated that the decoder’s output correlated with both variables, it was more strongly predicted by the relevant variable for multiplexed cells. The question of whether the choice information we observe is actively processed in auditory cortex or is a reflection of activity from another region is an interesting open question that goes beyond the scope of this study. We now say in the results: “Given the strong correlation between stimulus and choice variables in the task design, it is difficult to fully separate information about one variable from information about the other. […] This analysis establishes that a certain degree of separability is possible and demonstrates that the multiplexing observed in our decoding results is unlikely to be a trivial byproduct of correlations in the task variables.”

9) Title: Please consider revising your title to either omit or make more intuitive what "ensemble consensus" means and to clarify "nominally non-responsive", given the concerns of some of the reviewers about this terminology. One suggested revision "Response timing in frontal and sensory cortical neurons encodes task-relevant variables even when average responses are uninformative".

We have revised our title according to the reviewer’s suggestion. It now reads: “Spike-timing-dependent ensemble encoding by non-classically responsive cortical neurons.”

References:

Abbott LF1, Varela JA, Sen K, Nelson SB. Synaptic depression and cortical gain control. Science. 1997 Jan 10;275(5297):220-4. DOI: 10.1126/science.275.5297.221